# AWD regulates timed activation of BMP signaling in intestinal stem cells to maintain tissue homeostasis

Xiaoyu Tracy Cai[1], Hongjie Li[2], Abu Safyan[3,4,5], Jennifer Gawlik[4,5,6], George Pyrowolakis[4,5,7] & Heinrich Jasper [1,8,9]

Precise control of stem cell (SC) proliferation ensures tissue homeostasis. In the *Drosophila* intestine, injury-induced regeneration involves initial activation of intestinal SC (ISC) proliferation and subsequent return to quiescence. These two phases of the regenerative response are controlled by differential availability of the BMP type I receptor Thickveins (Tkv), yet how its expression is dynamically regulated remains unclear. Here we show that during homeostasis, the E3 ubiquitin ligase Highwire and the ubiquitin-proteasome system maintain low Tkv protein expression. After ISC activation, Tkv is stabilized by proteasome inhibition and undergoes endocytosis due to the induction of the nucleoside diphosphate kinase Abnormal Wing Disc (AWD). Tkv internalization is required for the activation of the Smad protein Mad, and for the return to quiescence after a regenerative episode. Our data provide insight into the mechanisms ensuring tissue homeostasis by dynamic control of somatic stem cell activity.

[1] Buck Institute for Research on Aging, 8001 Redwood Boulevard, Novato, CA 94945-1400, USA. [2] Department of Biology and Howard Hughes Medical Institute, Stanford University, Stanford, CA 94305, USA. [3] International Max Planck Research School for Molecular and Cellular Biology (IMPRS-MCB), Max Planck Institute of Immunobiology and Epigenetics, 79108 Freiburg, Germany. [4] Institute for Biology I, Faculty of Biology, Albert-Ludwigs-University of Freiburg, 79104 Freiburg, Germany. [5] Center for Biological Systems Analysis (ZBSA), Albert-Ludwigs-University of Freiburg, 79104 Freiburg, Germany. [6] Spemann Graduate School of Biology and Medicine (SGBM), Albert-Ludwigs-University of Freiburg, 79104 Freiburg, Germany. [7] Signalling Research Centre BIOSS and CIBSS, Albert-Ludwigs-University Freiburg, 79104 Freiburg, Germany. [8] Immunology Discovery, Genentech, Inc., 1 DNA Way, South San Francisco, CA 94080, USA. [9] Leibniz Institute on Aging - Fritz Lipmann Institute, 07745 Jena, Germany. Correspondence and requests for materials should be addressed to H.J. (email: jasperh@gene.com)

Effective tissue repair and regeneration in barrier epithelia relies on the dynamic control of stem cell (SC) activity[1–3]. In tissues like the mammalian airways and the *Drosophila* intestine, SCs are quiescent during homeostasis, but are rapidly and transiently activated in response to tissue damage[1,4,5]. The return to quiescence at the end of a regenerative episode is critical for tissue homeostasis, but remains poorly understood. *Drosophila* intestinal stem cells (ISCs) are rapidly activated in response to tissue damage to generate new enterocytes (ECs) and enteroendocrine (EE) cells. Several conserved signaling pathways, including JAK/STAT, EGFR, JNK, insulin, Hippo, Wingless, and BMP signaling, have been documented to stimulate ISC proliferation and self-renewal in such conditions[1,2,4,6–12]. Ca$^{2+}$ signaling integrates these various signals and sustains the proliferative response[13].

Mechanisms that promote ISC quiescence after a proliferative episode are less well understood, but include the targeted degradation of pro-proliferative transcripts by the mRNA degradation factor Tis11[14], as well as a secondary response to BMP-like ligands[15]. BMP signaling thus has complex roles in the control of ISC function, with several studies reporting contradictory consequences of BMP signaling perturbation in ISCs: long-term inactivation of Dpp signaling in both ISCs and EBs causes significant ISC loss by promoting symmetric non-self-renewing divisions[16], while short-term inhibition of Dpp signaling prevents DNA-damage induced proliferation of ISCs[17]. Upon *Erwinia carotovora carotovora 15* (*Ecc15*) infection, in turn, hemocyte-derived Dpp activates ISC proliferation to promote repair, while visceral muscle-derived Dpp inhibits ISC proliferation in the later phase of the response to promote ISC quiescence[15,18]. Dpp signaling also influences differentiation of midgut copper cells[19,20], and seems to be required for EC maintenance[21].

The complexity of the response of ISCs to the perturbation of Dpp signaling indicates that a detailed, temporally resolved characterization of Dpp signaling in ISCs and their daughter cells is critical. We have previously shown that the differentiation between a pro-proliferative and an anti-proliferative response of ISCs to Dpp is achieved by differential activation of the Type I receptors Saxophone (Sax) and Thickveins (Tkv), and of their downstream effectors Smad on X (SMOX), and Mad[15] (Fig. 1a). Sax is constitutively expressed in ISCs and responds to the early Dpp signal derived from hemocytes to promote ISC proliferation through SMOX, while Tkv is only detectable in ISCs in the later phase of the response. The presence of Tkv diverts the DPP response from Sax/SMOX signaling to MAD signaling, promoting a return of ISCs to quiescence. The dynamic regulation of Tkv expression thus functions as a key switch controlling the transition between activation and quiescence of ISCs, yet the mechanisms regulating the expression of Tkv in ISCs have not been resolved.

Here, we explore these mechanisms, and find that post-translational regulation of Tkv is critical for the dynamic control of Dpp responses during a regenerative episode. We find that Tkv turnover is regulated by the E3 ubiquitin ligase Highwire and by high proteasome activity in quiescent ISCs. In response to tissue damage, Tkv is temporarily stabilized due to general down-regulation of proteasome activity, and internalized into Rab5-positive endocytic vesicles. This internalization is facilitated by the *Drosophila* homolog of *Nm23* (*abnormal wing disks*, AWD), which is upregulated in active ISCs by JNK signaling. The AWD-facilitated endocytosis of Tkv is critical for the return of ISCs to quiescence, to prevent epithelial dysplasia, and for host survival during acute intestinal infection. Our findings identify a central mechanism responsible for a return to SC quiescence during regenerative processes, and has important implications for our understanding of SC regulation and tissue homeostasis in barrier epithelia.

## Results

**Tkv is induced and internalized in ISCs upon infection.** To explore the mechanisms regulating Tkv expression in ISCs, we generated a transcriptional reporter using the Tkv promoter (TkvA-lacZ; Supplementary Fig. 1a). Reporter activity was observed in ISCs both under homeostatic conditions and in activated ISCs (4 h post *Ecc15* infection; Fig. 1b), suggesting that the previously described induction of Tkv protein in the late phase of the regenerative response (determined using an antibody[15]), may be a consequence of post-transcriptional regulation of Tkv expression in ISCs. We performed qPCR analysis on FACS-sorted ISCs from flies exposed to *Ecc15* infection to further confirm this notion, and again found no significant changes in *tkv* mRNA levels in activated ISCs (12 or 18 h post *Ecc15* infection) compared to homeostatic conditions (Fig. 1c). To test directly whether Tkv expression is regulated posttranscriptionally, we used homologous recombination to generate a translational reporter line for Tkv which expresses a C-terminally tagged (3xHA) Tkv from its endogenous locus (Supplementary Fig. 1a)[22,23]. Tkv-3xHA localizes to the plasma membrane and distributes in a typical tissue pattern in 3rd instar larval imaginal disks (Supplementary Fig. 1b), and adult wings of Tkv-3xHA flies show no growth defects or other abnormalities (Supplementary Fig. 1c), suggesting that Tkv-3xHA flies are functionally wildtype. To confirm that the insertion of the 3xHA tag has no effect on gene expression and function of *tkv*, we knocked down *tkv* in wing imaginal disks, and found that TkvHA expression (Supplementary Fig. 1b, 1d) and pMAD (Supplementary Fig. 1d) levels were greatly reduced in imaginal disks. This was also repeated in ISCs after *Ecc15* infection (Supplementary Fig. 1e).

We crossed this line into animals containing a GFP lineage-tracing system initiated from ISCs ("escargot flip out", esg$^{ts}$ F/O[12]), and fed progeny *Ecc15* to induce a regenerative response. TkvHA expression was not observed in ISCs under homeostatic conditions, confirming the notion that its expression is regulated post-transcriptionally (Fig. 1d). Upon infection, however, ISCs upregulated TkvHA expression (Fig. 1d), recapitulating previous data from immunohistochemistry[15]. To assess the kinetics of this response, we examined expression of TkvHA in a time-course post *Ecc15* infection by immunohistochemistry. In contrast to Sax, which is continuously expressed under both homeostatic and infected conditions (Supplementary Fig. 1f), TkvHA was not observed in ISCs until 12 h post-*Ecc*15 challenge, and increased strongly around 18 h after challenge (Fig. 1e and Supplementary Fig. 1f). To confirm our immunohistochemistry quantification independently, we quantified TkvHA and Sax protein levels in sorted GFP-labeled ISCs by intracellular Flow Cytometry assay[24] every 6 h post-*Ecc15* challenge during a 24 h regenerative episode. These experiments confirmed the strong time-dependent induction of TkvHA in ISCs (up to 3-fold) upon *Ecc15* infection, with a peak around 24 h, further supporting a role for Tkv in the recovery phase after a regenerative response (Fig. 1f, Supplementary Fig. 7). Sax was continuously expressed and slightly upregulated in ISCs around 24 h (less than 2-fold), coinciding with the upregulation of TkvHA (Supplementary Fig. 1g). In the later phase of the proliferative response, Sax, and Tkv expression thus overlap, indicating that Tkv may have to compete with Sax for Type II receptor binding. We have shown previously that Sax/Smox signaling is repressed when Tkv/Mad signaling is activated, as indicated by the cytoplasmic localization of Smox and the phosphorylation of Mad (pMAD), respectively[15]. The dynamic regulation of Tkv in ISCs is independent from Sax, as loss of Sax or forced Sax overexpression did not prevent Tkv induction[15] (Supplementary Fig. 1e).

Our immunohistochemistry experiments also revealed that, in contrast to plasma membrane-bound Sax, Tkv has a distinct

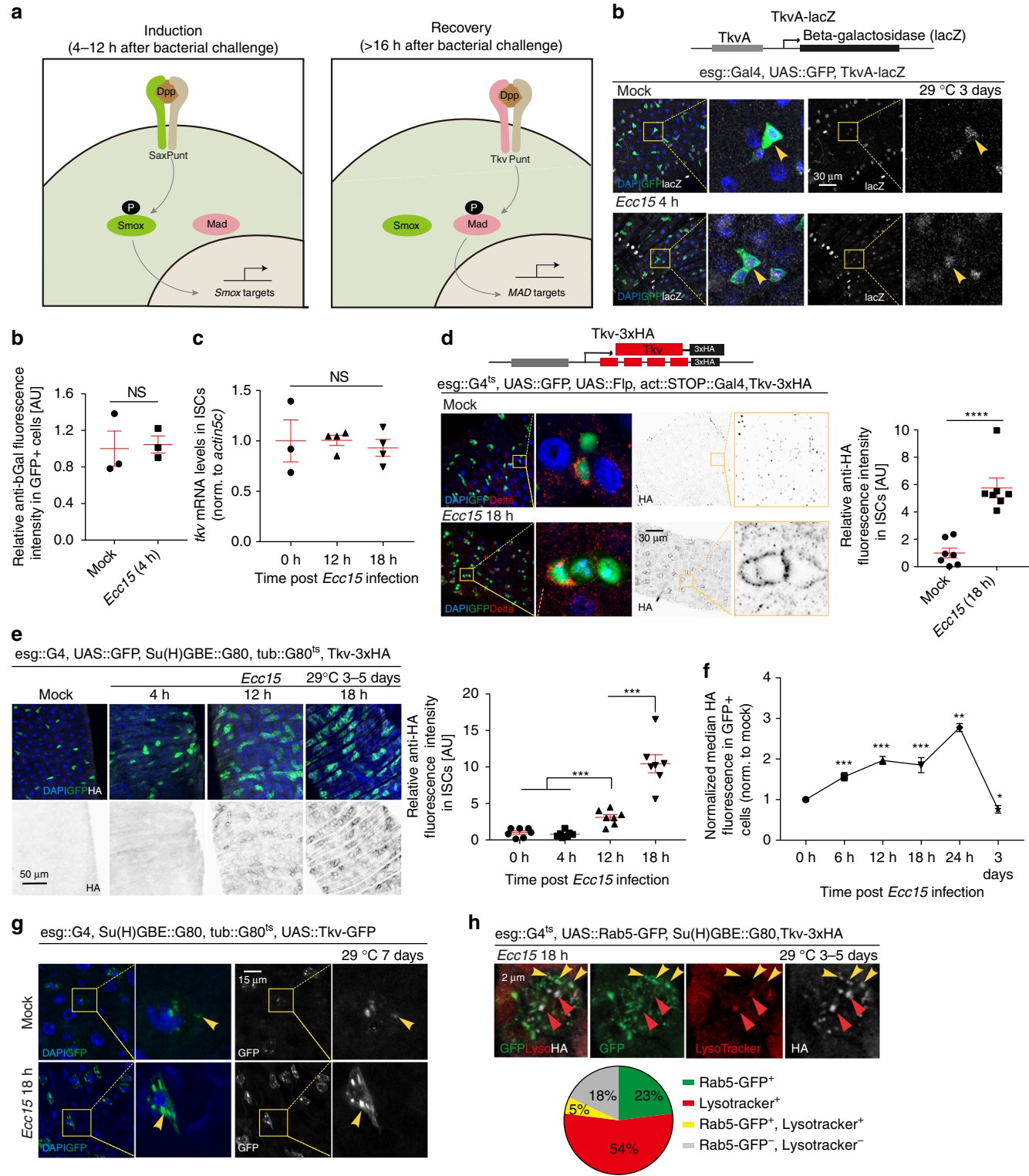

subcellular distribution in ISCs. Both overexpressed GFP-tagged Tkv (Tkv-GFP) and endogenous HA-tagged Tkv (TkvHA) localized to intracellular puncta in addition to its expected localization on the membrane (Fig. 1g, 1h and Supplementary Fig. 1f). These puncta corresponded to Rab5+ early endosomes and lysosomes (Fig. 1h), suggesting that Tkv is both stabilized and actively internalized during the late phase of a regenerative response. We confirmed the formation of Tkv-containing endocytic vesicles using fluorescence time-lapse imaging of ISCs over-expressing GFP-tagged Tkv

(Supplementary Movie 1). These observations are reminiscent of previously documented endocytosis of Tkv at larval neuromuscular junctions[25,26] and during wing posterior crossvein formation[27], and we decided to explore the ISC-specific mechanisms regulating the induction and endocytosis of Tkv during regeneration, as well as their functional consequences.

**Highwire and proteasome activity control Tkv turn-over.** Ubiquitin-mediated proteosomal degradation has been shown to

**Fig. 1** Tkv is induced and internalized in ISCs in response to *Ecc15* infection. **a** Model for the dynamic control of ISC activity by temporal regulation of Sax/Smox signaling and Tkv/Mad signaling in ISCs in response to infection or tissue damage. **b** Cartoon depicting the gene structure of TkvA-lacZ flies. Transcription from *tkv* promoter in ISCs (green, arrowheads) under homeostatic conditions (mock) and 4 h after *Ecc15* infection. **c** qPCR analysis of *tkv* mRNA level in ISCs under mock conditions and after *Ecc15* infection (normalized to *actin5c*). **d** Cartoon depicting the gene structure of Tkv-3xHA flies. Tkv-3xHA protein was detected within ISCs (esg::Gal4, UAS::GFP, green; Delta+, red) and ISC-derived daughter cells (GFP+, DELTA−), in posterior midgut (PM) of flies at 18 h after *Ecc15* infection, determined by immunohistochemistry with rabbit anti-HA antibody. Average HA fluorescence intensity in Delta+ cells was normalized to the mean value of mock. **e** Time-dependent induction of Tkv in ISCs, measured as the average intensity of Tkv-3xHA in ISCs per PM during the course of a 18 h *Ecc15* infection (normalized to the mean value of 0 h). **f** Median fluorescence of Tkv-3xHA in GFP + ISCs during the course of a 24 h *Ecc15* infection, as measured by intracellular Flow Cytometry analysis(normalized to the median value of control samples at 0 h collected on the same day of measurement). **g, h** Overexpressed Tkv-GFP fusion protein (**g**) or endogenous Tkv-3xHA protein (**h**) present as puncta in ISCs (GFP+). Tkv-3xHA puncta, detected by rat anti-HA antibody (used in the rest of our study), corresponded to Rab5+ early endosomes (green, yellow arrowheads) and lysosomes (red, red arrowheads), quantification of which were from single slice images (n = 101 cells from 7 guts) in **h**. Gain was increased for mock (**g**) for better visualization. Error bars indicated SEM (**b** n = 3 flies, **d** n = 7 flies, **e** n = 7 flies, **f** n = 447, 327, 3429, 1713, 1501, 1850, 1884, 2503 cells for 0 h samples; n = 2926, 2450, 1731, 727, 907, 792, 1049 cells for 6 h samples; n = 7230, 2880, 5792, 1209, 1539, 1798, 1636 cells for 12 h samples; n = 2901, 1466, 496, 943, 725, 891 cells for 18 h samples; n = 846, 1319, 1526, 725, 1490 cells for 24 h samples; n = 1148, 4492, 2098, 1954, 2357, 2542, 2666 cells for 3d samples). P values from Student's t-test (**b–d**, **e**) or from one-tailed Wilcoxon rank-sum test (**f**): ****P < 0.0001; ***P < 0.001; **P < 0.01; *P < 0.05; NS, not significant. Experiments were repeated 3 times (**d**, **e**, **g**, **h**)

influence turn-over of BMP receptors in multiple model systems[28–31], and we performed a limited genetic screen to identify possible regulators of Tkv stability in ISCs. We used RNA-mediated interference (RNAi) to target potential E3 ubiquitin ligases and protein kinases and other factors, including Highwire, Fused, Lkb1, Ube3a, Smurf, Dally, Dally-like (dlp), and Pentagone, to test whether these factors influence the kinetics of TkvHA expression in ISCs during regeneration (Supplementary Fig. 2a). This screen revealed that Highwire, a conserved RING-H2 E3 ubiquitin ligase[32–35], negatively regulates Tkv protein levels in ISCs. Highwire has been shown to interact with the Smad protein Medea (Med) to negatively regulate BMP signaling during the growth of neuromuscular synapses[32], but a role in Tkv protein turnover has not yet been reported. Flies with an ISC-specific knockdown of *highwire*, or carrying a mutation resulting in a large deletion of its N-terminal domain (including the RING finger domain required for E3 ubiquitin ligase activity; $hiw^{\Delta N/\Delta N})[33]$, exhibited increased Tkv on the surface of ISCs in homeostatic conditions (Fig. 2a, b and Supplementary Fig. 2b). However, these conditions did not influence downstream DPP signal transduction in ISCs, as assessed by measuring the levels of nuclear-localized pMAD (Fig. 2a, b and Supplementary Fig. 2b). To further test whether Highwire enzymatic activity is required for the regulation of Tkv stability in ISCs, we over-expressed High-wire$^{\Delta RING}$, a full-length Highwire with two point mutations in its RING finger domain, which have been reported to specifically disrupt its E3 ubiquitin ligase activity[33]. TkvHA was significantly stabilized in these ISCs (Fig. 2c), further supporting the notion that the enzymatic activity of Highwire is required for maintaining low Tkv expression in ISCs during homeostasis. We confirmed and quantified the stabilization of Tkv in ISCs using flow cytometry (Fig. 2e).

Inhibiting proteasome function by feeding flies the proteasome inhibitor PS-341 for 2 days resulted in similar ectopic expression of Tkv in ISCs without activation of pMAD (Fig. 2d, e), consistent with a role of the ubiquitin proteasome pathway in the degradation of Tkv in homeostatic conditions. We asked whether this observation was indicative of a general change in proteasome activity in activated ISCs, and used a CL1–GFP fusion protein[36], as a readout for proteasome activity in ISCs. CL-1 is a constitutive degradation signal that promotes rapid degradation of associated proteins by an active ubiquitin-proteasome system, and a GFP signal is thus only detectable in these cells when proteasome function is impaired. GFP levels increased significantly in ISCs as early as 4 h post *Ecc15* infection, and perdured until the late phase of the regenerative response, coinciding with Tkv accumulation (Fig. 2f). Using flow cytometry, we also quantified Highwire

protein levels after infection and found a slight upregulation at 12 h post-*Ecc15* challenge, but no significant changes at other time points (Fig. 2g), consistent with qPCR analysis of *highwire* mRNA in ISCs (Supplementary Fig. 2c).

Altogether, these results suggest that constitutive Highwire activity may license Tkv for degradation even in activated ISCs, but that inhibition of downstream proteasome activity prevents degradation and allows accumulation of Tkv in the late phase of the regenerative response.

**Infection induces AWD expression to promote Tkv endocytosis.** To investigate the mechanisms regulating the endocytosis of Tkv, we analyzed previously reported RNAseq data from ISCs isolated from Ecc15 infected animals[37]. We observed significant induction of the *Drosophila* homolog of the *Nm23* gene, *abnormal wing disks* (AWD), within 4 h after *Ecc15* infection, and this induction prevailed at 16 h after infection. A similar induction in ISCs isolated from infected intestines was also reported in a study published previously[38], suggesting that AWD is reproducibly induced during the activation of ISCs. We confirmed this induction using immunohistochemistry and flow cytometry, and found that AWD strongly accumulates in ISCs between 12 and 24 h post Ecc15 infection (Fig. 3a, b).

AWD, a nucleoside diphosphate kinase, generates GTP to support the function of dynamin (shibire) during synaptic vesicle internalization[39]. Dynamin is required for clathrin-mediated endocytosis[40], and, accordingly, for endocytosis of BMP receptors[25,27,41,42]. Previous genetic studies have suggested a role for AWD and its mammalian homolog Nm23 in the endocytic regulation of several cell surface proteins, such as platelet-derived growth factor/VEGF receptor (PVR)[43], FGF receptor[44], Notch[45], and Domeless[43]. To test whether AWD also facilitates Tkv internalization in activated ISCs, we overexpressed AWD in ISCs of infected flies. This resulted in larger Tkv puncta that corresponded to acidic compartments based on lysotracker staining (Fig. 3c, d, Supplementary Fig. 3b, c). Similarly, AWD overexpression was sufficient to promote the internalization of TKV in quiescent ISCs with repressed proteasome function or deficient in *highwire* (Supplementary Fig. 3d). AWD overexpression in homeostatic ISCs (where Tkv protein is not stabilized and TkvHA is not detectable), did not result in Tkv+ vesicles (Supplementary Fig. 3a), indicating that internalization of Tkv through AWD activity and stabilization of Tkv by down-regulation of proteasome activity are two separate processes that cooperate to regulate BMP signaling in ISCs. Highwire is not

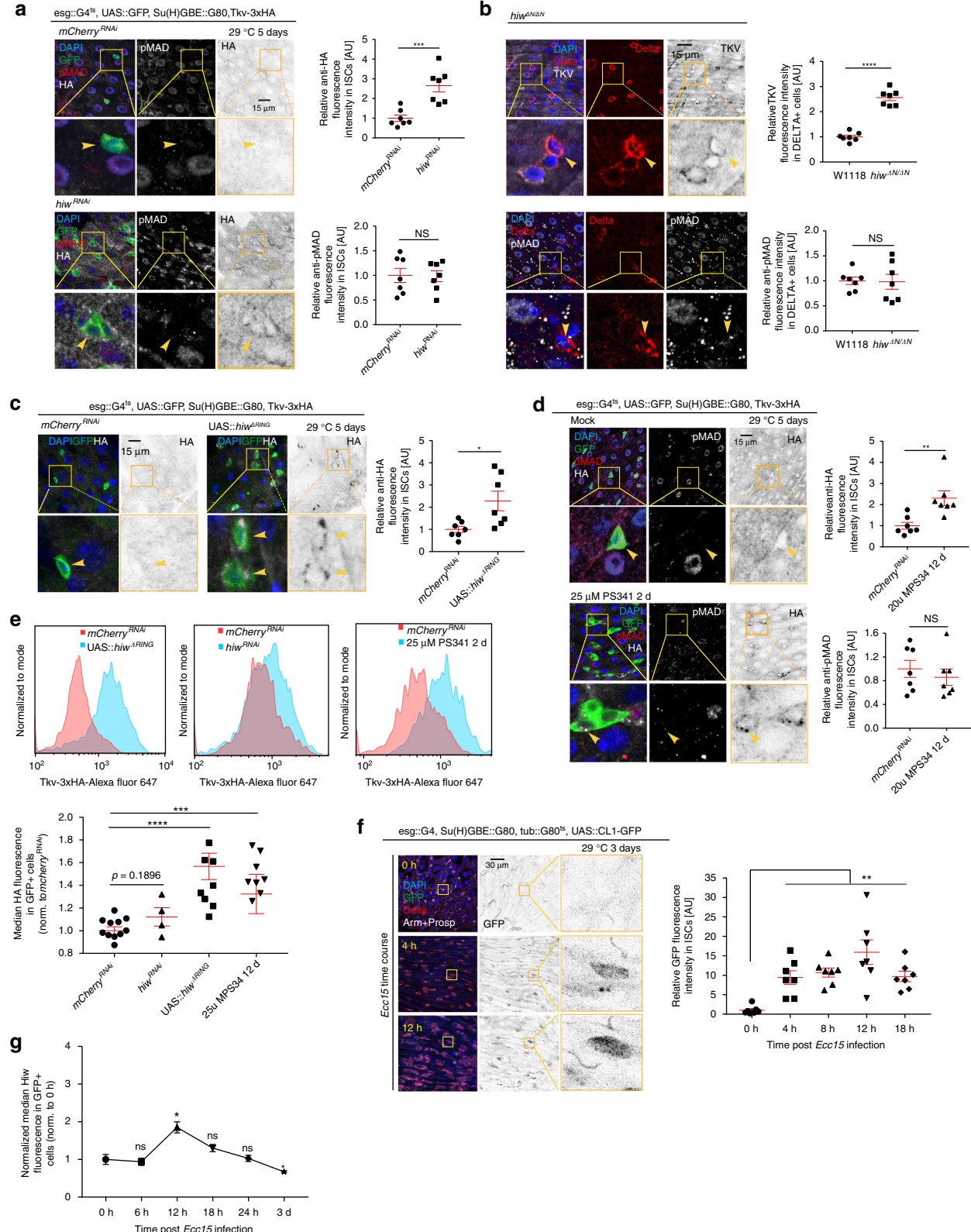

required for the internalization of TkvHA in AWD over-expressing ISCs (Supplementary Fig. 3d), further supporting this notion.

In turn, knockdown of AWD inhibited the internalization of Tkv after infection (Fig. 3e and Supplementary Fig. 3e; validation of the overexpression and knockdown efficiency are shown in Supplementary Fig. 4a). We confirmed this requirement for

AWD in Tkv internalization using mosaic analysis with a repressible cell marker (MARCM)[46] to generate ISCs homozygous for the *awd* loss of function allele *awd*[j2A4] [39,44] (Fig. 3f and Supplementary Fig. 4b). Two days after heat shock (AHS), and 18 h after *Ecc15* infection, Tkv accumulated on the membrane of AWD mutant (GFP+) cells, but was found in endocytic vesicles

**Fig. 2** Highwire and proteasome-dependent downregulation of Tkv during homeostasis. **a–d** Expression of Tkv-3xHA and phosphorylated MAD (pMAD) in wildtype ISCs (**a**, **c**, **d** arrowheads, GFP+; *mCherry*[RNAi] was expressed as RNAi control), *highwire*[RNAi] expressing ISCs (**a** arrowheads, GFP+), *highwire* [ΔN/ΔN] homozygous mutant ISCs (**b** arrowheads, Delta+), ISCs overexpressing *hiw*[ΔRing] (**c** arrowheads, GFP+), and ISCs in which proteasome activity was inhibited by 2 day feeding of 25 μM PS341 (**d** arrowheads, GFP+), respectively. Tkv-3xHA was detected by rat anti-HA antibody in **a**, **c**, **d**, while rabbit anti-TKV antibody was used in **b**. Quantifications of the average intensity of Tkv-3xHA and pMAD expression in ISCs per posterior midgut (PM) under the above conditions were normalized to the mean value of control samples respectively. The same quantifications for wildtype ISCs expressing *mCherry*[RNAi] were used in **a** and **d**, as both experiments were done at the same time. **e** Median fluorescence of Tkv-3xHA in GFP + ISCs under conditions as noted, measured by intracellular flow cytometry analysis. Fluorescence was normalized to the median value of control samples expressing *mCherry*[RNAi] collected on the same day of measurement. **f** Kinetics of ISC proteasome activity, measured as relative expression of CL1-GFP fusion protein in ISCs (Delta+) of PM during the course of an *Ecc15* infection episode of 18 h (normalized to the mean value of 0 h). Antibodies of Armadillo (Arm), labeling plasma membrane, and Prospero (Prosp), labeling enteroendoncrine cells, were used to help quantify GFP fluorescence intensity in ISCs. **g** Median fluorescence of Highwire in GFP + ISCs during the course of a 24 h *Ecc15* infection, as measured by intracellular Flow Cytometry analysis. Fluorescence was normalized to the median value of control samples at 0 h collected on the same day of measurement. Error bars indicate SEM (**a–d**, **f** n = 7 flies; **e** n = 3857, 3251, 4239, 4511, 80, 2123, 2657, 2246, 2000, 2237, 1515 cells for *mCherry*[RNAi] samples; n = 5730, 5567, 3793, 3765 cells for *hiw*[RNAi] samples; n = 6479, 7179, 6976, 6406, 3321, 3777, 4186, 1739, 2656, 2514 cells for UAS::*hiw*[ΔRING] samples; n = 4524, 5268, 5149, 3550, 1386, 918, 1305, 1269 cells for 25 μM PS341 2d samples, **g** n = 3086, 3887, 3390, 3908 cells for 0 h samples; n = 1770, 1580, 1710, 1694 cells for 6 h samples; n = 3148, 2623, 109, 2534 cells for 12 h samples; n = 4403, 3535, 4519, 4387 cells for 18 h samples; n = 5871, 5033, 5394, 4688 cells for 24 h samples; n = 4053, 4141, 4186, 3888 cells for 3d samples). *P* values from Student's *t*-test (**a–d**, **f**) or from one-tailed Wilcoxon rank-sum test (**e** and **g**): ****$P < 0.0001$; ***$P < 0.001$; **$P < 0.01$; *$P < 0.05$; NS, not significant. Experiments were repeated three times (**a–d** and **f**)

only in neighboring wild-type (GFP−) cells (Fig. 3f). The expression and localization of Sax, in contrast, was not affected by overexpression or knockdown of AWD in ISCs (Supplementary Fig. 4c).

We further monitored the dynamics of GFP-tagged Tkv in ISCs using time-lapse imaging. Over-expressing AWD increased the maximal number of Tkv puncta in ISCs and resulted in more Tkv puncta that co-localized with lysosomes under homeostatic conditions (Fig. 3g, h and Supplementary Fig. 4d, Supplementary Movie 2 and 3). Infection did not further increase these numbers, suggesting that induction of AWD in ISCs is the rate limiting step promoting endocytosis of Tkv after infection (Supplementary Fig. 4d, Supplementary Movie 4 and 5). Accordingly, knockdown of AWD significantly reduced average numbers of Tkv puncta in ISCs post septic challenge (Fig. 3g and Supplementary Fig. 4d, Supplementary Movie 6).

Combined, these data show that induction of AWD is sufficient and required for endocytosis of Tkv in ISCs.

**JNK regulates Tkv internalization through AWD.** To explore the upstream mechanisms responsible for inducing AWD and Tkv expression in ISCs, we tested whether candidate signaling pathways that are activated in ISCs upon septic injury and regulate ISC proliferation and differentiation may influence AWD and/or Tkv expression levels[1]. JNK signaling was both sufficient and required for AWD and Tkv induction in ISCs: knockdown of the JNK phosphatase Puckered (Puc), or overexpression of the JNK kinase hemipterous (Hep), induced the expression of Tkv and AWD in the absence of infection (Fig. 4a, b and e), while loss of the JNK basket (bsk) prevented induction in ISCs upon infection (Fig. 4c–e). We used MARCM clone analysis to confirm the requirement for *bsk*: *bsk* mutant ISCs failed to upregulate AWD and Tkv following *Ecc15* infection (Fig. 4f).

To test whether AWD acts downstream of JNK to regulate Tkv subcellular localization, we knocked down AWD in ISCs expressing *puc* RNAi. Loss of AWD inhibited the internalization of Tkv that was induced by JNK activation (Fig. 4g).

We also asked whether AWD overexpression was sufficient to promote the endocytosis of Tkv in *bsk* deficient ISCs, but since loss of JNK prevented Tkv accumulation in ISCs, its internalization could not specifically be assessed (Fig. 4h).

JNK thus promotes both stabilization of Tkv and AWD induction, while AWD functions to facilitate subsequent internalization of Tkv downstream of JNK.

**Internalization of Tkv optimizes BMP signal transduction.** While receptor endocytosis has been proposed to promote receptor turnover and downregulate signal transduction in many cases, it has also been found to be required to maintain or even promote signaling activity of specific receptors[27,47–51]. In Mv1Lu, R1B, and HepG2 cells, for example, transforming growth factor-β receptor (TGFβR) is internalized into sorting endosomes, where it phosphorylates its downstream transcription factor SMAD2[51]. The role of dynamin-dependent endocytosis in BMP signal transduction, however, remains controversial: while inhibiting endocytosis increases BMP-Smad signaling at the *Drosophila* neuromuscular junction[25], endocytosis has been found to promote signal transduction in neurons and the fly developing wing[27,52,53].

We asked whether AWD-facilitated endocytosis of Tkv is required for BMP signal transduction in ISCs, and found that over-expressing AWD in ISCs was sufficient to increase pMAD levels following *Ecc15* challenge (Fig. 5a). In addition, over-expression of AWD in wildtype flies fed PS-341 for 2 days (Supplementary Fig. 5a), or in ISCs deficient in *highwire* also further increased pMAD level in ISCs (Fig. 5c), suggesting that AWD increases BMP signaling activity in conditions in which Tkv protein is internalized. Accordingly, ISCs expressing two different RNAi constructs targeting AWD showed reduced pMAD levels after *Ecc15* infection (Fig. 5e), and AWD mutant ISCs did not up-regulate pMAD levels in response to *Ecc15* infection (Fig. 5g), confirming that AWD is required for BMP signal activation upon infection. These results were further confirmed using flow cytometry to quantify pMAD levels in GFP-labeled ISCs (Fig. 5b, d, f).

The GTPases Rab5, Rab7, and Rab11 play important roles in sorting/early endosomes, late endosomes, and recycling endosomes, respectively, during endocytosis. Since Tkv puncta preferentially localized to Rab5+ early endosomes upon infection, and further overexpression of AWD after septic challenge promoted Tkv co-localization with lysosomes, we wondered whether the AWD-facilitated endocytic regulation of BMP signal transduction is dependent on Rab5. We knocked down Rab5, while over-expressing AWD in ISCs following *Ecc15* challenge for 18 h, and found that, while Tkv aggregated on the ISC membrane in these conditions, pMAD levels were not increased (Supplementary Fig. 5b).

Since AWD has been shown to be required for the endocytosis of Notch[45], and since we found that, similar to *notch* deficient ISCs, *awd* mutant ISCs fail to properly differentiate (Supplementary Fig. 4b), we asked whether loss of *notch* affects the localization of TkvHA and activation of MAD. Loss of Notch did not prevent the

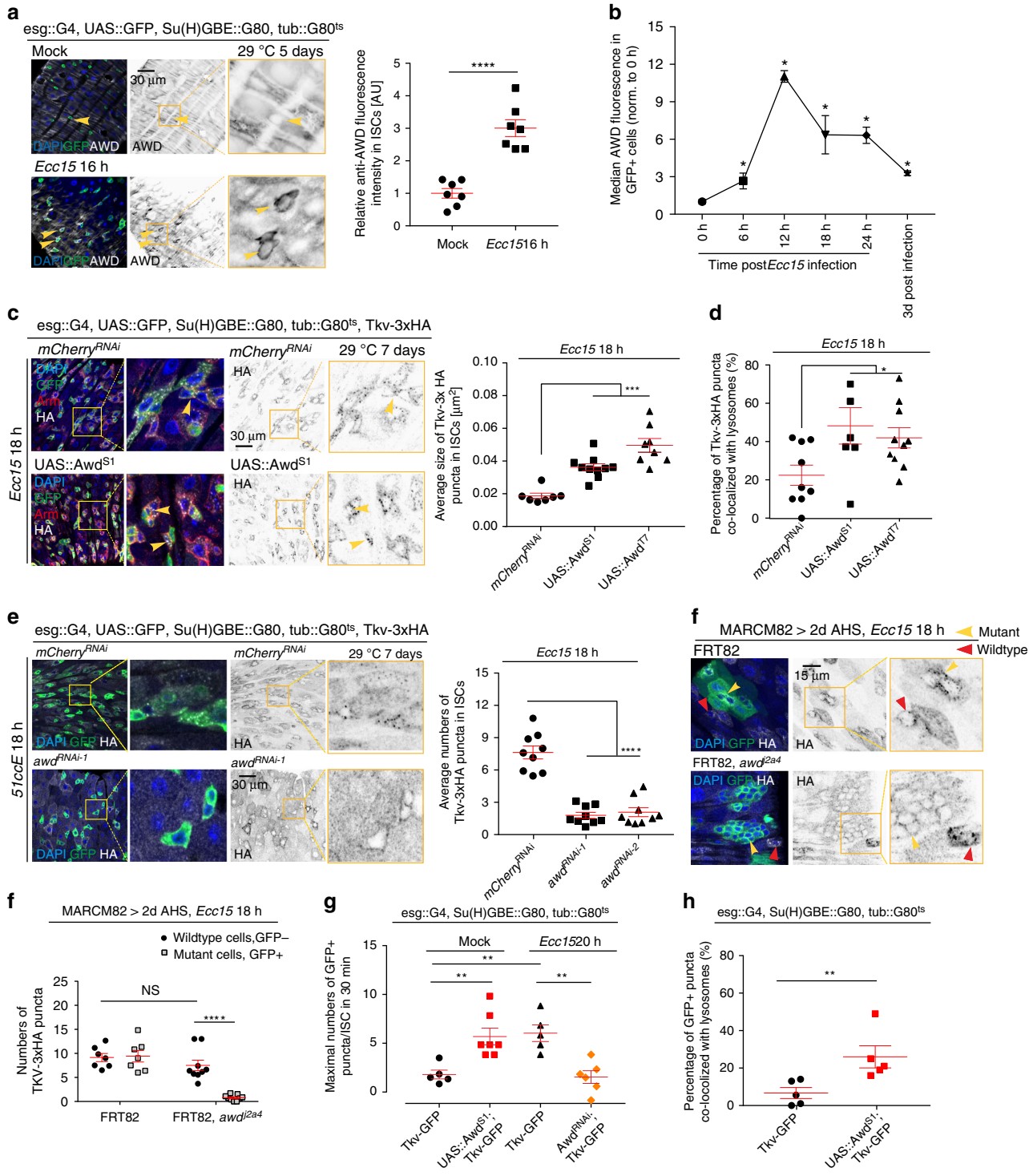

endocytosis of TkvHA or MAD phosphorylation (Supplementary Fig. 5c), indicating that the endocytic regulation of Dpp signaling by AWD is independent of Notch signaling.

Since AWD provides GTP for dynamin, we also asked whether the regulation of BMP signal transduction by AWD in ISCs depends on dynamin. Indeed, loss of dynamin (shibire) inhibited the induction of pMAD in AWD overexpressing ISCs (Supplementary Fig. 5b).

Altogether, these results confirm that AWD acts to facilitate Tkv internalization through dynamin-mediated endocytosis in ISCs, and that this internalization is required for the activation of BMP signaling in the recovery phase during intestinal regeneration.

**AWD/Tkv/MAD restore ISC quiescence during regeneration.**
To test whether, similar to Tkv[15], AWD is required to restore ISC quiescence and re-establish tissue homeostasis after regeneration has concluded, we measured the number of mitotic figures (phospho-histone H3 + ISCs) in a time-course following infection with Ecc15. Loss of Tkv or MAD prevented ISC recovery to quiescence post infection, as reported (Supplementary Fig. 6a). Overexpression of AWD did not further inhibit ISC proliferation after recovery from Ecc15 challenge (Supplementary Fig. 6b), but knockdown of AWD prevented re-establishment of ISC quiescence, resulting in ISC over-proliferation at 24 h post Ecc15 infection (Fig. 6a), similar to Tkv or

                                        

**Fig. 3** AWD promotes internalization of Tkv. **a** AWD protein level in ISCs (GFP+) with or without *Ecc15* infection determined by immunohistochemistry. **b** Median fluorescence of AWD in GFP + ISCs during the course of a 24 h *Ecc15* infection, as measured by intracellular Flow Cytometry analysis. Fluorescence was normalized to the median value of control samples at 0 h collected on the same day of measurement. **c** Tkv-3xHA puncta (arrowheads) in wildtype or *awd* overexpressing ISCs (GFP+) after 18 h of *Ecc15* infection were revealed by immunohistochemistry. Sizes of Tkv-3xHA puncta were quantified. **d** Co-localization between Tkv-3xHA puncta and lysosomes in wildtype or *awd* overexpressing ISCs after 18 h of *Ecc15* infection was quantified. **e** Tkv-3xHA expression in wildtype or *awd*[RNAi] expressing ISCs upon 18 h of *Ecc15* infection. Average number of Tkv-3xHA puncta in ISCs per posterior midgut (PM) was quantified. **f** Differential Tkv-3xHA expression patterns inside (GFP+, indicated by yellow arrowhead) and outside (GFP−, indicated by red arrowhead) of *awd* mutant clones after *Ecc15* challenge. Numbers of Tkv-3xHA puncta was quantified. **g** Maximal numbers of Tkv-GFP puncta within 30 min of time-lapse imaging were quantified in ISCs in which *awd* was overexpressed in the absence of infection, or after 20 h of *Ecc15* infection in ISCs in which *awd* was knocked down. **h** Quantification of the percentage of Tkv-GFP puncta co-localized with lysosomes in wildtype and *awd* overexpressing ISCs in the absence of infection, from 30 min time-lapse movies. Error bars indicate SEM (**a** n = 7 flies, **b** n = 3086, 3887, 3908 cells for 0 h samples; n = 1770,1710,1694 cells for 6 h samples; n = 3148, 2623, 109, 2534 cells for 12 h samples; n = 4403, 3535, 4519, 4387 cells for 18 h samples; n = 5871, 5033, 5394, 4688 cells for 24 h samples; n = 4053, 4141, 4186, 3888 cells for 3d post infection samples, **c** n = 7–9 flies; **d** n = 6–10 flies; **e** n = 9 flies; **f** n = 7–9 flies; **g, h** n = 5 flies). P values from Student's t-test (**a, c–h**) or from one-tailed Wilcoxon rank-sum test (**b**): ****P < 0.0001; ***P < 0.001; **P < 0.01; *P < 0.05; NS, = not significant. Experiments were repeated three times (**a, c–f**)

MAD loss of function conditions (Supplementary Fig. 6a). Loss of AWD also led to intestinal dysplasia, while wildtype flies fully recovered from *Ecc15* challenge after 2 days (Fig. 6b). Over-expressing GFP-tagged wildtype Tkv inhibits ISC proliferation in wildtype animals[15], but we did not observed this when ISCs were deficient in AWD (Fig. 6c), in line with our observation from live imaging that loss of AWD reduces the number of TKV-GFP puncta in ISCs (Fig. 3g, Supplementary Movie 4 and 6). Even 2 days after regeneration concludes in wild-type flies, AWD deficient ISCs continued to be active (Fig. 6b). Overexpressing constitutively active Tkv, in turn, was sufficient to rescue ISC over-proliferation caused by AWD knockdown at 24 h post septic challenges (Fig. 6d).

The induction of Tkv endocytosis by AWD is thus required for the re-establishment of ISC quiescence. We asked whether this role for AWD would also influence the maintenance of barrier function of the intestinal epithelium during regeneration. We used the "Smurf" assay, in which intestinal penetration of a blue food dye is assessed, to measure barrier function[54], and assessed resilience to infection in animals orally infected with *Pseudomonas Entomophila (PE)*, an enteropathogen that induces tissue damage and causes death. Flies lacking Tkv/MAD or AWD showed increased barrier dysfunction and rapidly succumbed to *PE* infection (Fig. 6e, g), suggesting that recovery of ISC quiescence by inducing AWD and activating Tkv/MAD signaling plays a critical role in host survival to infection. Overexpression of constitutively active Tkv, however, also caused increased mortality (Fig. 6h), and, accordingly, does not rescue barrier dysfunction or mortality after acute *PE* infection in AWD knockdown conditions (Fig. 6f, h). It is likely that this is due to continuous inhibition of ISCs proliferation by constitutively active Tkv (Supplementary Fig. 6c), preventing proper regeneration of the epithelium.

## Discussion

Our study provides insight into the regulation of BMP/Dpp signaling in ISCs during regenerative episodes. While previous studies had described the activation of Tkv/MAD signaling in ISCs as an essential step to limit proliferation in the late phase of a regenerative response[15–17,21], it remained unclear how Tkv expression and MAD activation were controlled during that phase. Our data identify Highwire-mediated Tkv degradation as a mechanism to maintain low Tkv expression in quiescent ISCs, as well as changes in proteasome activity and subsequent AWD-facilitated internalization of Tkv as essential steps for the activation of MAD signaling. We further connect this mechanism to JNK-mediated activation of AWD expression, thus providing a temporally resolved model for the transition from activated ISCs to resting ISCs after regeneration has concluded (Fig. 7). We find that this control of Tkv internalization in the late phase of the

regenerative response is essential for the re-establishment of epithelial homeostasis after injury (Fig. 7).

Although previous studies have indicated that clathrin/dynamin-dependent endocytosis of BMP type I receptors is conserved in various models, including *C. elegans*, *Drosophila*, mouse, and human fibroblasts, the role of endocytosis in BMP signal transduction remains inconclusive, and may depend on the specific endocytic trafficking routes of the receptors in each context[25–27,48,49,55,56]. Our data suggest that the stabilization of Tkv, and its accumulation on the plasma membrane is not sufficient to activate MAD signaling in ISCs, but that AWD-facilitated internalization into Rab5+ endosomes is required. We did not observe a significant decline in TkvHA levels in *awd* gain of function conditions, but our live imaging results revealed that increased *awd* activity can promote localization of Tkv to lysosomes, suggesting that AWD-facilitated receptor internalization promotes both signaling activity and the subsequent turnover of ligand receptor complexes in lysosomes.

The endocytic role of AWD has been implicated in the regulation of multiple signaling pathways during development, including platelet-derived growth factor/VEGF receptor (PVR)[43], FGF signaling[44], Notch signaling[45], and Domeless signaling[45]. Our results suggest that in ISCs, AWD specifically regulates Tkv/MAD signaling, and that this regulation is independent of its role in Notch signaling, as the internalization of Tkv and the activation of pMAD is not affected by loss of *notch* in ISCs.

AWD participates in endocytosis by enhancing dynamin (shibire) activity and by regulating Rab5 function in early endosome maturation[43–45,57,58]. Accordingly, MAD phosphorylation downstream of AWD-facilitated endocytosis of Tkv is Rab5 and dynamin-dependent in ISCs. This supports the role of AWD as a supplier of GTP for Rab5 and dynamin GTPases, and contributes to our understanding of the complex role of endocytosis in BMP signal transduction. Dynamin-dependent endocytosis has been shown to negatively regulate BMP-Smad signaling at *Drosophila* neuromuscular junction[25], but this contrasts with its function in neurons and wing crossvein formation[27,52]. Our data suggest positive regulation of BMP signaling by endocytosis in ISCs.

Previous studies have suggested that ISC proliferation is regulated by a complex interplay of cell autonomous and non-autonomous signals, which engage JNK and Dpp signaling, as well as other signaling pathways, including JAK/STAT, EGFR, Insulin, Hippo, and Wingless signaling[1,2,4,6–12,54]. A sophisticated understanding of signaling dynamics controlling the transition from quiescent to activated ISCs after tissue damage is emerging[4,13]. Our study clarifies the molecular events that mediate the return from the activated state to quiescence in the recovery phase of the regenerative response. JNK activation, which occurs early in regenerative responses[59], initiates this transition by

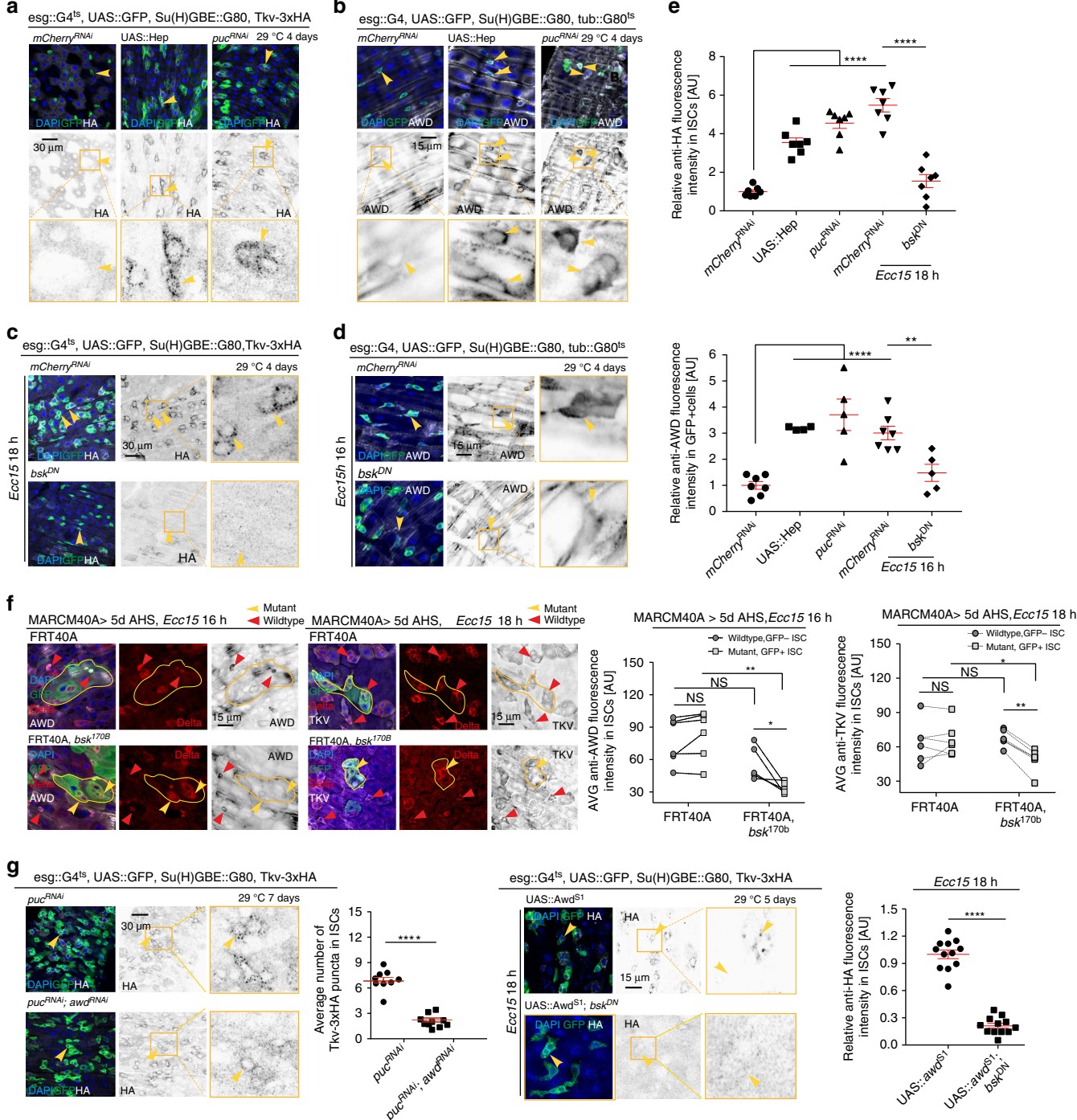

**Fig. 4** JNK signaling promotes Tkv internalization through AWD in ISCs. **a**, **b** Expression of Tkv-3xHA (**a**) or AWD (**b**) in ISCs (arrowheads, GFP+) determined by immunohistochemistry, when JNK was activated by overexpressing JNK Kinase (*hep*) or knocking down *puckered* (*puc*) for 4 days. **c**, **d** Expression of Tkv-3xHA (**c**) and AWD (**d**) in ISCs (arrowheads, GFP+) upon 18 h *Ecc15* infection, when JNK signaling was deactivated by overexpressing a dominant negative form of JNK (*bsk*DN) in ISCs. **e** Normalized average intensity levels of Tkv-3xHA and AWD in ISCs per posterior midgut (PM) under conditions of **a**–**d** (normalized to the mean values of control samples expressing *mCherry*RNAi). **f** Differential expression of AWD or TKV protein between *bsk* mutant ISCs (DELTA+, GFP+, indicated by yellow arrowheads) and wildtype ISCs (DELTA+, GFP−, indicated by red arrowheads) after *Ecc15* challenge. Lines connect data for wildtype and mutant ISCs from same guts in quantification. **g** Tkv-3xHA expression in *puc*RNAi ISCs (arrowheads, GFP+) with and without *awd* loss of function. Tkv-3xHA punctum numbers were quantified. **h** Expression level of Tkv-3xHA in *awd* overexpressing ISCs with and without deactivating JNK signaling by co-expressing *bsk*DN, measured as the average HA intensity in ISCs after 18 h *Ecc15* infection. Quantifications were normalized to the mean values of samples overexpressing *awd* alone. Error bars indicate SEM (**e** n = 7 flies, **f** n = 5–7 flies, **g** n = 9–11 flies, **h** 12 flies). *P* values were calculated from Student's *t*-test, except that ratio paired *t*-test was performed to examine the significant differences of AWD or TKV expression between wildtype ISCs and *bsk*170b ISCs within the same gut (**f**): ****P < 0.0001; **P < 0.01; *P < 0.05; NS, not significant. Experiments were repeated 3 times (**a**–**e**, **g**–**h**) or twice (**f**)

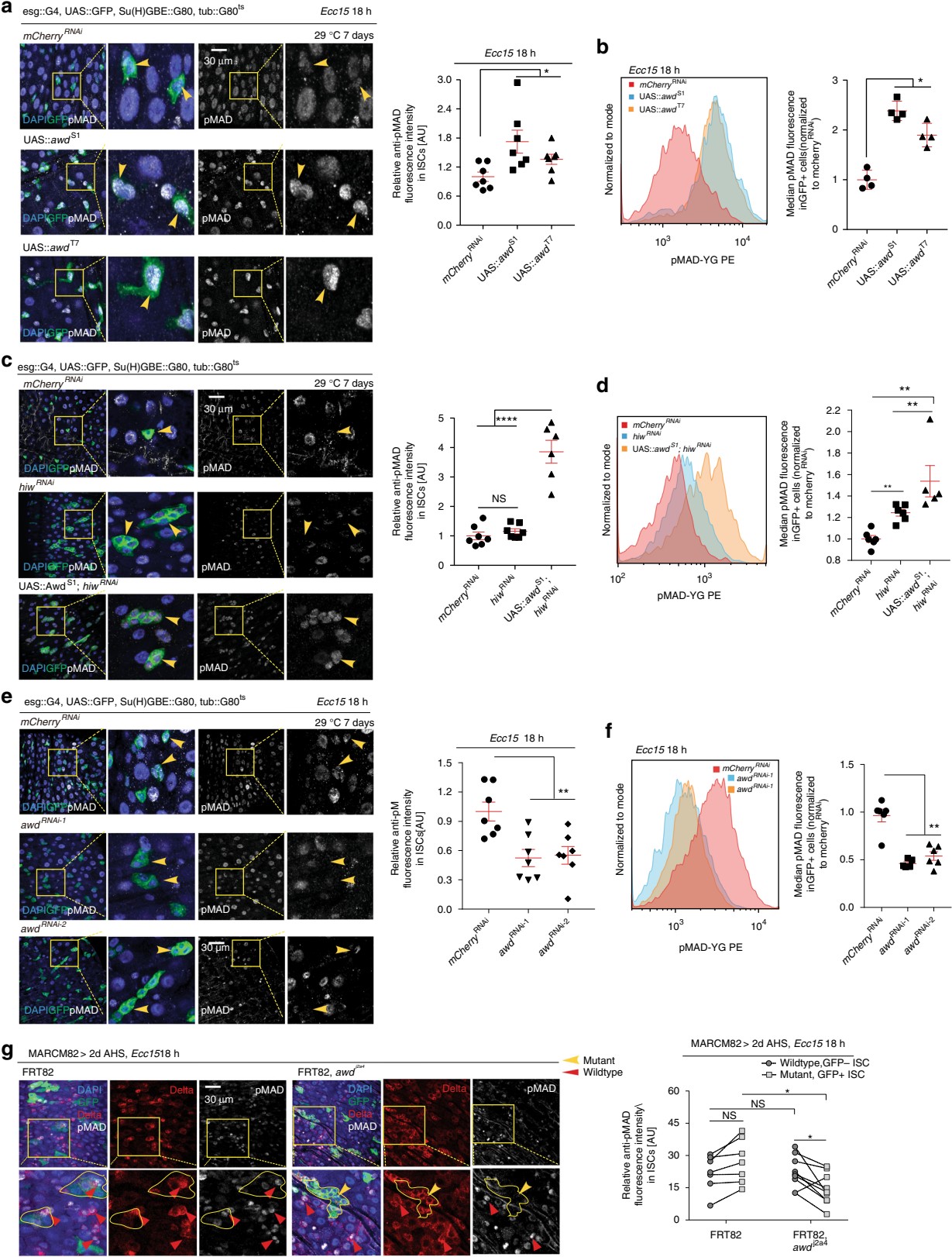

promoting Tkv stabilization and internalization. Detailed in vivo analysis of signaling pathway activities for other pro-proliferative pathways in ISCs during the regenerative response will be needed to address the question of whether specific inactivation of those pathways in ISCs also contributes to the return to quiescence.

Dpp signaling has a complex role in the regulation of intestinal homeostasis[15–18,21], and our data support previous findings reporting that Tkv/MAD signaling is required for the re-establishment of epithelial homeostasis in the recovery phase of the regenerative response. Accordingly, lack of AWD/Tkv/MAD

**Fig. 5** AWD regulates MAD signaling in ISCs. **a, e** pMAD level in wildtype (**a** and **e** arrowheads), *awd* over-expressing (**a** arrowheads) and *awd*[RNAi] (**e** arrowheads) ISCs (GFP+) at 18 h after *Ecc15* challenge determined by immunohistochemistry. Average intensity levels of pMAD in ISCs per posterior midgut (PM) were quantified and normalized to the mean value of control samples expressing *mCherry*[RNAi]. **c** Expression of pMAD in *highwire*[RNAi] ISCs (arrowheads, GFP+) with and without *awd*[OE] was revealed by immunohistochemistry. **b, d, f** Histogram overlay of pMAD fluorescence in GFP+ ISCs under conditions as noted, measured by intracellular flow cytometry assay. Median fluorescence intensity of pMAD in GFP+ ISCs under these conditions, was computed by FlowJo software and normalized to the median value of control samples expressing *mCherry*[RNAi] collected on the same day of measurement. **g** Differential expression of pMAD between *awd* mutant ISCs (DELTA+, GFP+, indicated by yellow arrowheads) and wildtype ISCs (DELTA+, GFP−, indicated by red arrowheads) after 18 h of *Ecc15* challenge. Lines connect data for wildtype and mutant ISCs from same guts in quantification. Error bars indicate SEM (**a** $n = 7$ flies, **b** $n = 2529, 2919, 2815, 2294$ cells for *mCherry*[RNAi] samples; $n = 1923, 3162, 3359, 3551$ cells for UAS::*awd*[S1] samples; $n = 16,325, 16,655, 15,328, 16,586$ cells for UAS::*awd*[T7] samples, **c** $n = 6$–7 flies, **d** $n = 713, 743, 782, 1521, 2106, 2056$ cells for *mCherry*[RNAi] samples; $n = 973, 1256, 1516, 1325, 3765, 3793$ cells for *hiw*[RNAi] samples; $n = 233, 425, 473, 442, 2087$ cells for UAS::*awd*[S1];*hiw*[RNAi] samples, **e** $n = 7$ flies, **f** $n = 10416, 6072, 6729, 10756, 9809, 10,267$ cells for *mCherry*[RNAi] samples; $n = 20866, 16933, 19561, 8922, 9318$ cells for *awd*[RNAi-1] samples; $n = 9880, 15,935, 10,604, 8867, 9808, 6798$ cells for *awd*[RNAi-2] samples, **g** $n = 6$–7 flies). *P* values were calculated from Student's *t*-test in **a, c, g**, and **e**, except that the ratio paired *t*-test was performed to examine the significant differences of pMAD expression between wildtype ISCs and *awd* mutant ISCs within the same gut in **g**. One-tailed Wilcoxon rank-sum test was performed in **b, d** and **f**. ****$P < 0.0001$; **$P < 0.01$; *$P < 0.05$; NS, not significant. Experiments were repeated 3 times (**a, c**, and **e**), or twice (**g**)

---

signaling leads to ISC over-proliferation, epithelial dysplasia, and barrier dysfunction. These findings are consistent with the role of BMPR in the mammalian intestinal epithelium regeneration, as loss of BMPRIA causes overgrowth of the crypt[60].

Which downstream mechanisms mediate these effects of Tkv/ MAD signaling, however, remains largely unknown. A previous study from our lab has identified Tis11 as a critical mediator of the return to quiescence in ISCs[14]. Tis11 promotes degradation of mRNAs encoding pro-mitotic factors, and its expression peaks around 16 h post *Ecc15* infection, coinciding with Tkv accumulation and MAD activation. It is therefore interesting to speculate that Tis11 acts downstream of MAD signaling to promote the re-entry into quiescence. Future studies will investigate this hypothesis.

To date, studies of AWD using model organisms have provided important insight into its role in vesicle transport during development[43–45,57,58]. AWD, or its mammalian homolog NME1, has also been identified as a conserved metastasis suppressor by regulating tumor cell motility and invasion[58,61–63]. Our study contributes to this body of work by providing evidence supporting an essential role of AWD in the control of ISC proliferative plasticity and epithelial regeneration. It is expected that additional downstream effectors of AWD beyond Tkv/MAD signaling (such as changes in Notch signaling activity) contribute to the precise control of ISC function and intestinal regeneration. Given the evolutionary conservation of the investigated signaling pathways, as well as of the control of somatic stem cell regulation[64], it will be of interest to investigate this role of AWD/NME1 in tissue regeneration further. New insight into the control of tissue regeneration and homeostasis can be expected from such work.

## Methods

***Drosophila* stocks, husbandry, and treatments**. Flies were kept on standard fly food at 25 °C and 65% humidity with a 12 h light/dark cycle and only female animals were used in all experiments. The TARGET system was used altogether with the indicated Gal4 driver to conditionally express UAS-linked transgenes. Flies were maintained at 18 °C on standard fly food and 2–6-day-old female adults were transferred to 29 °C for 4–7 days to induce genetic expression, unless otherwise indicated.

The following fly lines were obtained from Bloomington *Drosophila* Stock Center: w[1118], *mcherry* RNAi(35785), UAS::Tkv-EGFP (51653)[15], *smox* RNAi (26756)[15], *mad* RNAi (31315)[15], *hiw* RNAi(28031), *awd* RNAi#1(33712), *awd* RNAi#2(42532), *shibire* RNAi(28513), *rab5* RNAi(30518), UAS::Rab5-GFP(43336), *bsk*[DN] (6409), *fused* RNAi(31043), *ube3a* RNAi(31972), *smurf* RNAi(40905). *pentagone* RNAi (51169), *dally* RNAi (28747), *dally-like(dlp)* RNAi (34089), UAS:: Tkv[QD] (36536).

The following fly lines were provided by Vienna *Drosophila* RNAi Center: *tkv* RNAi (3059), *puc* RNAi (3018), *lkb1*RNAi(108356KK).

The following fly lines were gift from other labs: UAS::CL1-GFP(Dr. Udai B Pandey), Hiw ^ΔN/ΔN^ (Dr. Arson DiAntonio), UAS::Awd S1 and UAS::Awd T7 (Dr. Tien Hsu[44]), FRT82, Awd[j2a4] (Dr. Tien Hsu), UAS::Hep (Dr. Marek Mlodzik),

MARCM40A (hsFlp;FRT40A tub-Gal80;tub-Gal4,UAS-GFP) (Dr. Benjamin Ohlstein), FRT40A, bsk[170b](Dr. Nicholas E. Baker), *notch* RNAi and MARCM82 (hsFlp; tub-Gal4, UAS-GFP; FRT82, tubGal80) (Dr. Nobert Perrimon), UAS:: hiw^ΔRing^ (Dr. Pejmun Haghighi), esg[ts]F/O (esgGal4, tubG80[ts], UAS-GFP; UAS-flp, act > STOP > Gal4, Dr. Huaqi Jiang).

UAS::Sax(F001576) was obtained from FlyORF(Zurich ORFeome Project).

Standard fly food was prepared with the following recipe: 1 L distilled water, 22 g molasses, 6.2 mL propionic acid, 13 g agar, 80 g corn flour, 65 g malt extract, 18 g brewer's yeast, 10 g soy flour, 2 g methyl-p-benzoate in 7.3 mL of EtOH. For proteasome inhibitor feeding experiments, 20 μM Bortezomib (PS-341) from ApexBio(Cat No.A2614) was additionally added. For Smurf experiments, 500 mg/mL of Blue dye no. 1 (Alfa Chem) was additionally added, and 45–60 female flies (20–30 flies per vial) were flipped three times a week. Smurf flies were counted visually every other day.

This study follows all ethical regulations for research required for the use of *Drosophila melanogaster* as an animal model. Complying with NIH regulations, no ethical approval was required for work with *Drosophila melanogaster*.

**Generation of TkvA-lacZ and Tkv-3xHA reporter lines**. TkvA is an intronic fragment from the *tkv* genomic locus (around 4 kb, Supplementary Fig. 1a) and was amplified by genomic PCR using primers: BglII_tkvA (forward): GGTTTa-gatctAGGATCAGAGGGATATGAGGATGCC; Acc65I_tkvA (reverse): GGTTTggtaccGACGAATGTGCAACAGTTGGAAACGC. The fragment was cloned into the BglII/Acc65I sites of the reporter vector placZattB and the construct was inserted in the landing site attP2 on chromosome 3 L by phiC31/attB integration. TkvA was the only fragment from a collection of reporter constructs tilling the genomic locus of *tkv* to activate reporter expression in ISCs.

Genome engineered *tkv* carries three tandem repeats of the hemaglutinin (HA) tag at the C-terminus and was generated by homologous recombination followed by site-directed insertion[22,23]. First, the two last exons of the gene were replaced by an attP-containing cassette using the vector pTV(Cherry) and homology arms flanking the deleted segment, generating *tkv*[ko,attP]. In the second step, the gene was reconstituted by re-inserting the missing exons to generate *tkv*[attP/B,3xHA] (*tkv*-3xHA in short). To this end, the last two exons of *tkv* (including the intervening intron) were cloned into the vector RIV(white), modified to include sequences coding for the HA tags just prior to the stop codon and inserted in the *tkv*[ko,attP] chromosome by standard phiC31/attP transgenesis. The mini-white cassette used for selection was removed by Cre-mediated recombination resulting in the final *tkv*-3xHA chromosome, which, besides the HA tag, contains a attP/B hybrid sequence in the last large intron of *tkv* and a single loxP site in the 3'UTR of the gene as remnants of the genome engineering. The integrity of the generated chromosomes was confirmed molecularly and genetically. Flies with *tkv*-3xHA as the only source of TKV develop normally, display no obvious phenotypic abnormalities and are fully fertile (Supplementary Fig. 1b, c).

**Immunostaining**. Adult female *Drosophila* guts were dissected in 1×, PH 7.4 phosphate-buffered saline (PBS), and fixed for 30 min at room temperature in fixation buffer containing: 25 mM KCl, 100 mM glutamic acid, 1 mM MgCl₂, 20 mM MgSO₄, 4 mM sodium phosphate, and 4% formaldehyde. Guts were washed for 30 min—1 h at 4 °C in washing buffer containing: 1× PBS, 0.5% bovine serum albumin and 0.1% Triton X-100, followed by incubation in primary antibodies overnight at 4 °C, 1 h washing at 4 °C, and secondary antibodies for 2 h at room temperature. For pSMad3 staining, phosphatase inhibitor (Roche, 4906837001) was added in fixation buffer, 1 h wash, and primary antibody incubation following the same protocol above. For Delta staining, methanol–heptane fixation method was used[20]. For LysoTracker staining, guts were dissected following the above protocol and incubated in 1× PBS containing 50 μM

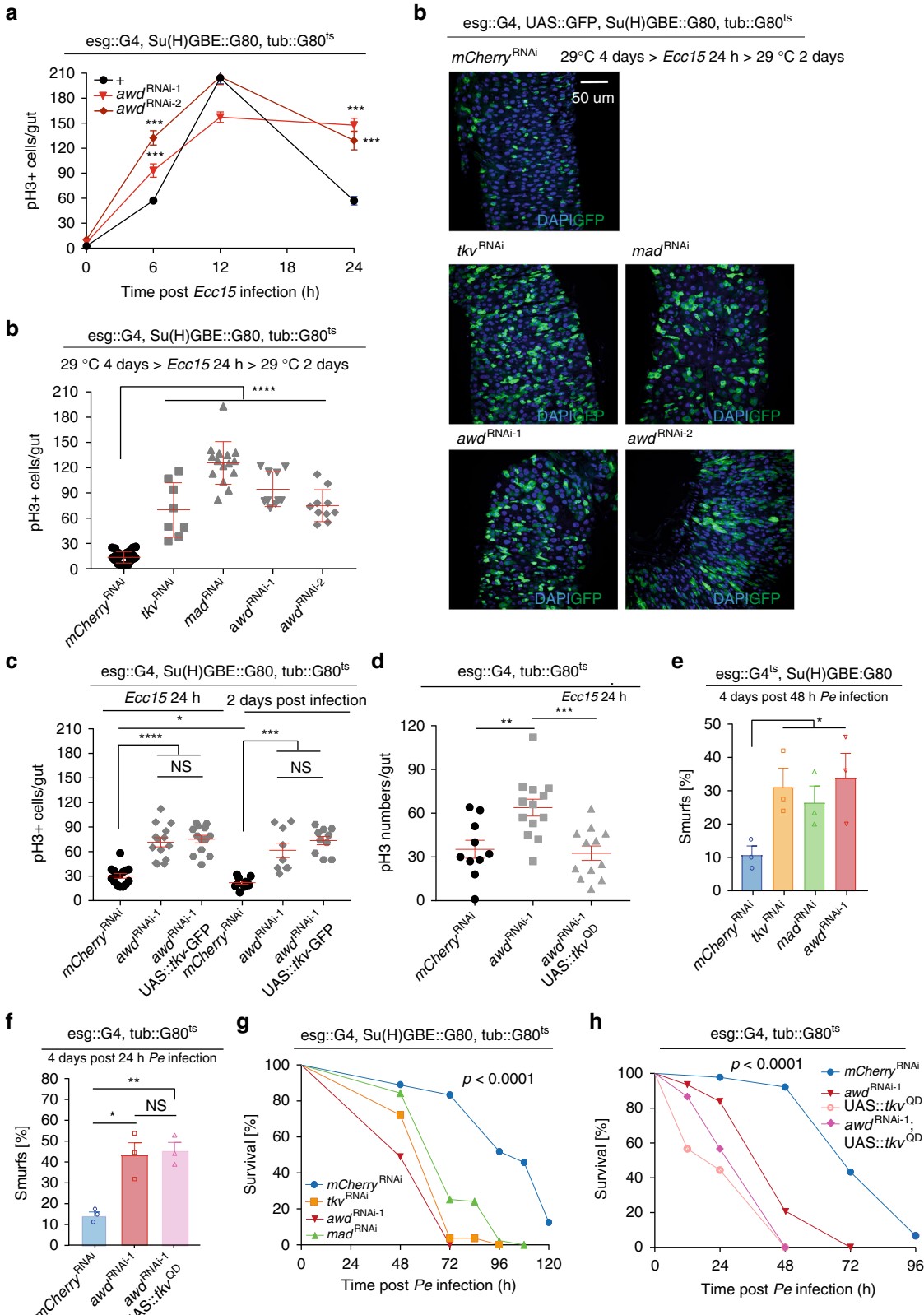

LysoTracker® Red DND-99 (Thermo Fisher Scientific, L7528) for 5–10 min at room temperature. Guts were washed in 1× PBS for three times (10 min each time), followed by regular fixation protocol as described above.

Primary antibodies and dilution used in this study: rabbit anti-pSMad3 (Epitomics Cat. No. EP823Y, 1:500), rabbit anti-β-galactosidase (Cappel MP Biomedicals, 1:5000), rabbit anti-phospho-Histone H3 Ser 10 (EMD Millipore Cat. No. 06–570,

1:1000), rabbit anti-AWD (gift from Dr. Tien Hsu, 1:100), mouse anti-Armadillo (DSHB, 1:100), mouse anti-Prospero (DSHB, 1:50), rabbit anti-Sax (Abcam Cat. No. ab42105, 1:200), rat anti-Delta (gift from Dr. Matthew D. Rand, 1:1000), rabbit anti-Tkv (gift from Dr. Marcos Gonzalez-Gaitan, 1:100), rat anti-HA (Roche Cat. No. 11867423001, dissolved in distilled water and stored at 100ug/ml, 1:300), rabbit anti-HA (Cell Signaling Cat. No. 3724S, 1:100). Fluorescent secondary antibodies were

**Fig. 6** AWD/Tkv/MAD promote ISC quiescence and host resistance to acute infection. **a** Dynamic mitotic activity of ISCs when *awd* was specifically knocked down with two different RNAi lines was measured as numbers of phospho-Histone H3+ (pH3+) cells per gut during the course of an *Ecc*15 infection episode of 24 h. **b** Gut dysplasia was observed at posterior midgut 2 days after *Ecc*15 oral infection when *tkv*, *mad*, or *awd* were knocked down in ISCs respectively. Numbers of pH3+ cells per gut of these conditions were quantified. **c**, **d** Quantification of ISC mitotic activity in response to *Ecc*15 infection, when *awd* was knocked in ISCs with and without co-overexpressing Tkv-GFP fusion protein (**c**) or constitutively active Tkv (Tkv$^{QD}$, **d**), measured as numbers of pH3+ cells per gut. **e** Portion of Smurf flies when *tkv*, *mad* or *awd* were knocked down in ISCs respectively, monitored after a prior feeding on *PE* for 48 h. **f** Portion of Smurf flies when *awd* was knocked down in ISCs and EBs (with esgG4, tubG80$^{ts}$ driver) with and without Tkv$^{QD}$ co-overexpression, monitored after a prior feeding on *PE* for 24 h. **g** Survival rates of flies (same as **e**) in response to acute intestinal damage were monitored after continuous *PE* infection. **h** Survival rates of flies (same as **f**) in response to acute intestinal damage were monitored after continuous *PE* infection. Error bars indicate SEM (**a** $n = 10$–17 flies, **b** $n = 8$–29 flies, **c** $n = 9$–15 flies, **d** $n = 10$–13 flies, **e** $n = 45, 50, 45$ flies for *mCherry*$^{RNAi}$ samples; $n = 45, 50, 47$ flies for *tkv*$^{RNAi}$ samples; $n = 60, 50, 51$ flies for *mad*$^{RNAi}$ samples; $n = 50, 50, 46$ flies for *awd*$^{RNAi-1}$ samples, **f** $n = 89, 86, 88$ flies for *mCherry*$^{RNAi}$ samples; $n = 80, 73, 56$ flies for *awd*$^{RNAi-1}$ samples; $n = 58, 61, 61$ flies for *awd*$^{RNAi-1}$; UAS::*tkv*$^{QD}$ samples). *P* values in **g** and **h** were calculated from log rank test (cohort sizes: **g** $n = 40, 53, 42, 81$ flies for *mCherry*$^{RNAi}$, *tkv*$^{RNAi}$, *awd*$^{RNAi}$, *mad*$^{RNAi}$, respectively; **h** $n = 90, 62, 90, 60$ flies for *mCherry*$^{RNAi}$, *awd*$^{RNAi}$, UAS:Tkv$^{QD}$, *awd*$^{RNAi}$; UAS:Tkv$^{QD}$, respectively). Other *P* values from Student's *t*-test: ****$P < 0.0001$; ***$P < 0.001$; **$P < 0.01$; *$P < 0.05$, NS, not significant. One representative image from 4 to 7 flies tested in a single experiment was shown in **b**. Experiments were reproduced twice (**a**–**c**, **g**), or three times (**d**, **h**)

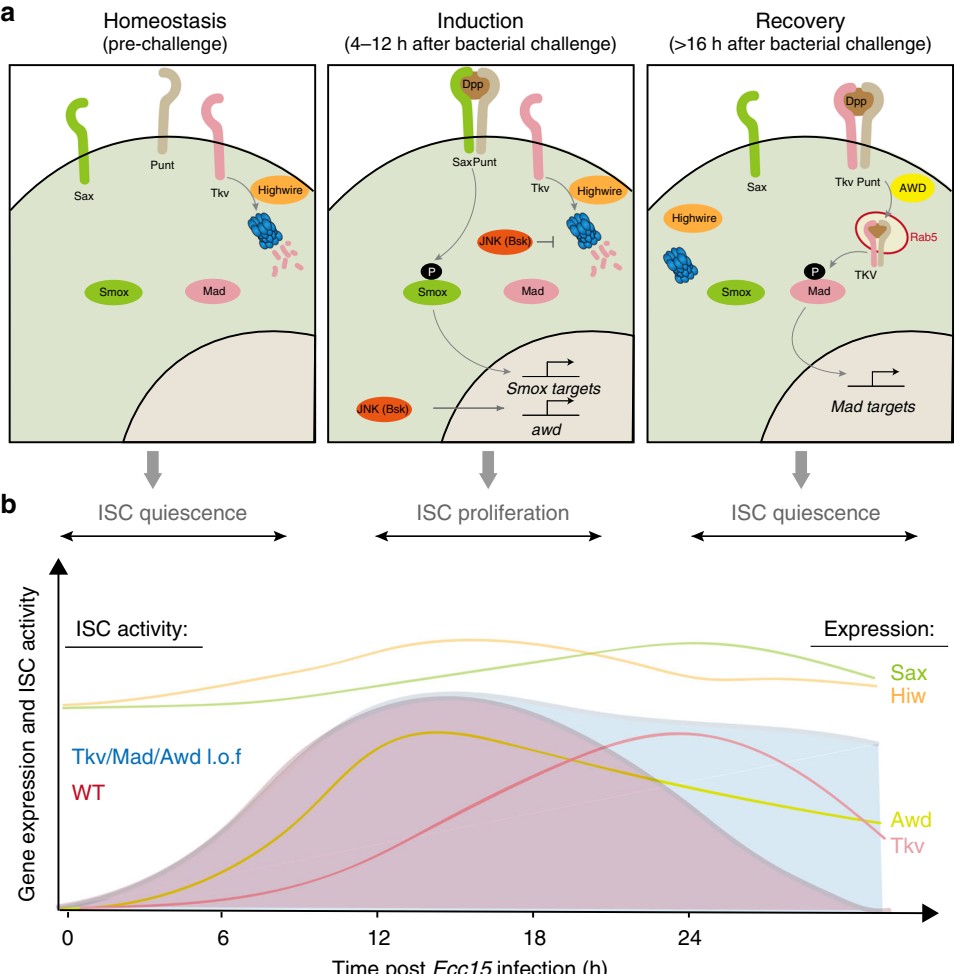

**Fig. 7** Model. **a** Model for the dynamic control of ISC activity by AWD-facilitated endocytic regulation of Tkv/MAD signaling during the regenerative response. Under homeostatic conditions, Highwire and proteasomes maintain high turn-over of Tkv protein, leading to its absence in ISCs. During the early induction phase after infection, Dpp ligands first bind to constantly expressed Sax, promoting ISC proliferation through Smox signaling. Meanwhile, activated JNK stabilizes Tkv on ISC membrane, potentially by inhibiting proteasomal activity, while upregulating AWD expression. During the late recovery phase, stabilization of Tkv protein followed by AWD-facilitated internalization to early endosomes switches on MAD signaling, allowing a return to ISC quiescence. **b** Timeline of ISC proliferation and relative expression of Tkv/Sax/Highwire/Awd during one regeneration episode

bought from Jackson Immunoresearch. DAPI was used to stain DNA. All the images were taken on a Zeiss LSM 700 confocal microscope or on the Yokogawa CSU-W1/ Zeiss 3i Marianas spinning disk confocal microscope using 20× or 40× objective, except that images in Fig. 1h were taken by Leica SP5 confocal microscope using 100× objective. All the images were processed by Illustrator and ImageJ.

**Time-lapse live imaging of *Drosophila* intestines.** Adult female flies were dissected in Shields and Sang M3 insect medium (Sigma Cat. No. S8398). Intestines were transferred to a 35 mm glass bottom dish (MatTek, P35G-1.5–14-C), with 50–100 μl 3.5% low melting agarose (dissolved in M3 insect medium) added on top. After 20 min, 3 mL M3 insect medium was added in the dish, and intestines were imaged at

intervals of 30 s for 20–30 min on a Zeiss LSM 780 confocal microscope using 40× objective. For LysoTracker labeling, intestines were transferred to 1× PBS containing 1 μM LysoTracker® Red DND-99 (Thermo Fisher Scientific Cat. No. L7528) after dissection and incubated at room temperature for 5–10 min followed by washing with 1× PBS for 20 min. Movies were analyzed using Image J.

**Cell sorting and RT-qPCR.** Flies expressing cytosolic GFP specifically in ISCs were crossed to $mCherry^{RNAi}$. 70–100 intestines were dissected per condition in cold dissection buffer (1× PBS, 1% BSA, 5% FBS) and treated with 0.1% trypsin for 1 h at 29 °C, followed by pipetting up-and-down to dissociate tissue. Cells were collected after centrifugation at 4 °C, 500×g for 5 min, and resuspended in dissection buffer. GFP+ cells were sorted followed by RNA extraction with Trizol (Invitrogen). cDNA synthesized by using an oligo-dT primer was then applied in real-time PCR on a Bio-Rad CFX96 detection system with the following primers, or on a QuantStudio 8 Flex system (ThermoFischer) with the following Taqman Probes (ThermoFischer). For data analysis, C(t) values of hiw or tkv levels in linear scale were normalized to actin5c.

Primer sequences are listed as following:
hiw (F): 5′-CACGCGCAGAAAAATGCAAC-3′;
hiw (R): 5′-CCGCATTCCCTTCCAGAACA-3′;
actin5C (F): 5′-CTCGCCACTTGCGTTTACAGT-3′;
actin5C (R): 5′-TCCATATCGTCCCAGTTGGTC-3′.
Taqman probes are listed as following:
tkv(Dm01844694_g1); actin5c(Dm02361909_s1).

**Intracellular flow cytometry and FACs analysis statistics.** Fifteen to twenty female fly guts were dissected for each biological replicate. Single cell suspension of each sample was prepared freshly, following the above dissociation step by trypsin. Cells were immediately transferred to 1.2 mL FACs tubes (ThermoFisher Cat. No. 3487), washed with 1× cold PBS, and resuspended with fixable viability dye eFluor™ 780 solution (1:1000, diluted in 1× cold PBS). After 20 min incubation on ice, cells were washed twice with 1× cold PBS and spun down at 4 °C, 300×g for 5 min. Samples were pulse vortexed for 10 s in residual volume of remains (around 50–100 μL), to completely dissociate the pellet before fixation.

eBioscience™ Foxp3/Transcription Factor Staining Buffer Set (ThermoFisher Cat. No. 00–5523–00) was used for the following fixation and permeabilization steps. Cell pellet was resuspended in 500 μL fixation buffer (1 part of fix/ permeabilization concentrate: 3 parts of diluents), and fixed at room temperature for at least 30 min. Fixed cells were spun down at 700×g for 5 min and blocked at room temperature for 10 min in blocking buffer (1× permeabilization buffer containing 2% goat serum). For antibody staining, cells were incubated at room temperature for 30 min–1 h with primary antibodies diluted in blocking buffer. Cells were then washed with 1× permeabilization buffer twice and incubated with fluorescent secondary antibodies diluted in blocking buffer at room temperature for 30 min–1h. Cells were washed with 1× permeabilization buffer twice and resuspended with 1× FACs buffer (1× PBS, 0.5% BSA, 0.05% Na Azide). DAPI was added to each sample at a final concentration 1 μg/mL, to stain nuclei, and analyzed by BD Symphony flow cytometer.

Primary antibodies and dilution used in this study: rabbit anti-pSMad3 (Epitomics Cat. No. EP823Y, 1:800), rat anti-HA (Roche Cat. No. 11867423001, dissolved in distilled water and stored at 100 μg/mL, 1:500), rabbit anti-Sax (Abcam Cat. No. ab42105, 1:500), rabbit anti-AWD (gift from Dr. Tien Hsu, 1:300), mouse anti-Highwire (DSHB, 6H4, 1:300). Fluorescent secondary antibodies were bought from Jackson Immunoresearch (1:500).

FlowJo v10 Software computed the median fluorescence intensity (MFI) of channels of interest in GFP-labeled ISCs, and generated histogram (x-axis: fluorescence intensity levels of channels of interest in logarithmic scale; y-axis: the number of events, noted as modal). To overlay multiple cell populations with different sizes, the absolute cell counts were normalized to the peak height at mode of the distribution, noted as normalized to mode in y-axis. To combine values from experiments conducted on different days, all values collected on the same day were normalized to the median value of control samples on the same day of measurement. Median value of fluorescence minus one control (FMO) containing all the fluorochromes except for the one being measured was subtracted from the median values of all the samples in Fig. 1f and Supplementary Fig. 1g, to reduce the background. One-tailed Wilcoxon rank-sum test was used to compare the significant differences of MFIs between samples by Prism.

**MARCM clone induction.** Two to three-days-old flies were heat-shocked for 45 min at 37 °C and kept at room temperature for 2–5 days as indicated, followed by bacterial infection before dissection.

**Bacterial infection.** Bacterial strains, Ecc15 or Pseudomonas entomophila (PE), were cultured in LB medium at 30 °C for 20–24 h. Bacteria were centrifuged at 5000 rpm for 10 min at room temperature and resuspended in 500 μL 5% sucrose (OD100) for the time indicated. Flies were starved in empty vials for 2 h before transferred to vials containing bacteria. For Smurf assay, flies were infected with PE for 1 or 2 days, as indicated, and shifted to normal Smurf food, followed by visually

counting numbers of blue flies. For survival experiments, flies were infected with PE continuously, and 100 μL 5% sucrose was added every day to vials until all the flies died.

**Image quantification and statistical analyses.** Confocal images were obtained using a Zeiss LSM 780 confocal microscope, Zeiss 3i Marianas spinning disk confocal microscope and Leica SP5 confocal microscope. Images to be compared were collected using identical laser and detector settings unless noted otherwise, and analyzed using NIH ImageJ. Quantifications were only performed in the posterior midgut. Mean values of background signal in the vicinity of the analyzed ISCs were subtracted from absolute fluorescence intensity values of individual ISCs of interest, followed by normalization to the mean of control samples.

Sample size and number of replicates are described in the corresponding legends. Statistical analyses were performed with Prism (GraphPad Software, La Jolla, CA, USA) and MS Excel software. Student's t-test was used to compare means from two independent groups of data with normal distribution, except that ratio paired t-test was used to compare mean values of AWD, Tkv, and pMAD expression between mutant ISCs and wildtype ISCs within the same gut in MARCM clone analysis in Figs. 4f and 5d. Log-rank test was used to test for statistical significance in survival assay. No statistical method was used to predetermine sample sizes, and no randomization was performed among differentially treated animals.

**Reporting summary.** Further information on research design is available in the Nature Research Reporting Summary linked to this article.

## Data availability
The authors declare that the data supporting the findings of this study are available within the paper and its supplementary information files.

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

## Acknowledgements

We thank Drs. Tien Hsu, Udai B Pandey, Arson DiAntonio, Marek Mlodzik, Benjamin Ohlstein, Nicholas E. Baker, Nobert Perrimon, Huaqi Jiang, Matthew D. Rand, Pejmun Haghighi, and Marcos Gonzalez-Gaitan for flies and reagents. We thank Dr. Daniel Jun-kit Hu for the help in ex vivo live imaging and Leica SP5 confocal microscopy. Work in the H.J. lab was supported by NIH RO1s AG050104, AG047497, and GM117412. Work in the GP lab was supported by the German Research Foundation (DFG) under Germany's Excellence Strategy (BIOSS-EXC294, CIBSS—EXC-2189—Project ID390939984) and PY72/2–1.

## Author contributions

X.C. and H.J. designed all experiments. X.C. performed all experiments and analyzed data. G.P., A.S., and J.G. generated TkvA-LacZ and TKV-3xHA fly lines and validated their expression accuracy (Supplementary Fig. 1a–d). H.L. screened Tkv-lacZ promoter lines (Fig. 1b). X.C. and H.J. wrote the manuscript.

## Additional information

**Competing interests:** The authors declare no competing interests.

