## [Peer Review File · Nature Communications]

Reviewers' comments:

Reviewer #1 (Remarks to the Author):

In *Drosophila*, intestinal regeneration induced by injury involves two steps: 1) early activation of intestinal stem cell (ISC) proliferation and 2) subsequent return to homeostatic status. It has been shown previously that differential regulations of the BMP type I receptor Tkv is critical for these two steps of the regenerative process, however, the underlying molecular mechanism remains unclear. In the manuscript, Cai et al showed that the E3 ubiquitin ligase Highwire and proteasome maintain Tkv protein at low levels during homeostatic condition. They further found that upon Ecc15 infection, Tkv was temporarily stabilized and internalized into Rab5-positive endocytic vesicles. Moreover, they found that the internalization of TKV in ISCs is regulated by AWD, which is upregulated by Ecc15 infection. Finally, the authors performed extensive experiments to show that the JNK/AWD/TKV regulatory axis is involved in the re-establishment of ISC quiescence.

Overall, this work are of potential interest and would provide insights in understanding how the precise regulation of Tkv contributes to intestinal regeneration. It is worth to note that the manuscript includes two parts: 1) describing the role of Highwire and proteasome in maintaining Tkv protein at a low level under homeostatic condition and 2) investigating functional contribution of AWD to Tkv internalization and the re-establishment of ISC quiescence. However, two parts do not come together in a cohesive way, and interpretations of some data are confusing. Thus, additional biochemistry experiments elucidating the molecular mechanisms of how TKV stability is regulated by Highwire in homeostatic condition and how Tkv is activated by AWD during internalization are necessary in a revised manuscript.

Below are specific comments:

1. Regulation of Tkv by Highwire is not clear, since all conclusions were made from immunostaining experiments, levels of endogenous TKV mediated by Highwire and proteasome in guts under homeostatic or infective conditions should be examined by western blotting assays.
2. As seen from several figures, levels of Tkv were strongly induced in guts upon infection of Ecc15 or by overexpression of AWD. Authors explained that these changes were mainly attributed to the inhibition of proteasome activity. The transcriptional reporter of TkvA used in the work is a transgene that may not fully recapitulate the behavior of endogenous tkv. The reviewer suggest that authors perform qPCR to measure levels of tkv transcript under the investigated conditions in a time-course way.
3. In abstract, authors claimed that "Tkv protein levels are kept low during homeostasis by the activity of the E3 ubiquitin ligase Highwire and degradation by the ubiquitin-proteasome system". However, authors did not provide any evidence showing that Highwire targets and regulates Tkv through ubiquitination and that enzymatic activity is required for Tkv ubiquitination. The reviewer suggest that authors measure levels of ubiquitinated Tkv in guts with/without knockdown of highwire and in guts treated with/without proteasome inhibitor PS-341. Since the authors claimed "Inhibiting proteasome function by feeding flies the proteasome inhibitor PS-341 for 2 days resulted in similar ectopic expression of Tkv in ISCs without activation of pMAD (Fig. 2c)", the reviewer want to see if there is difference in levels of ubiquitinated Tkv between gut samples with highwire knockdown and samples treated with proteasome inhibitor.
4. Since Highwire and AWD were proposed to regulate Tkv dynamics during homeostasis and early ISC activation, respectively, the major question the reviewer wants to ask is what relationship between Highwire and AWD is. It has been proposed that the receptor internalization is often associated with its ubiquitination. Given that AWD promotes Tkv internalization, thus possibly promoting Tkv ubiquitination, the question becomes as to whether Highwire contributes to the AWD-mediated Tkv internalization. Authors should discuss this issue, if Highwire has no role in the later stage of gut regeneration.

Reviewer #2 (Remarks to the Author):

The study entitled "AWD regulates BMP Signaling in intestinal stem cells to maintain tissue homeostasis" identifies Highwire-mediated Tkv degradation and AWD mediated internalization of Tkv as essential steps for the activation of MAD signaling in ISCs. They further link this mechanism to JNK-mediated activation of AWD expression, thus providing a model for the transition from activated ISCs to resting ISCs after regeneration has concluded. This is an exciting story elucidating relevant pathways regulating ISC and tissue homeostasis. While the manuscript and the experiments described therein provide a wealth of experimental data, critical evidence is still missing to support major conclusion and many important questions remain.

Major comments

1, Lack of statistical analysis for critical results:

Lack of quantification in Fig. 1b, Fig. 1c, Fig. 2a-c, Fig. 4a-d, and Fig. 5a-c.

This reviewer is not convinced by the data shown that Tkv protein, but not transcripts of tkv, is affected by Ecc15 infection.

Co-localization of Rab5, Tkv, and Lysosome in Fig. 1f also needs to be quantified based on images from single slice or a few slice projections. Co-localizations analysis on images from thick tissues could lead to false positive results.

LysoTracker labels acidic organelles such as lysosome; this might explain the co-localization between lysosome and the Rab5-positive early endosomes.

2, Image quality needs to be improved throughout. For example, the resolution for anti-pMAD staining is too low in Fig. 2a and 2c. The anti-pMAD staining pattern in controls (control RNAi and mock-treated) is greatly different without any explanation. Inconsistent patterns of anti-HA staining are also noted in Fig. 1C and 1F in similar conditions.

Proper controls are needed in many places. For example, Delta immunostaining is needed to label ISC in Fig. 3e, as used in Fig. 4e and 4f. There is obvious AWD expression in GFP+ and GFP- cells in Fig 3a Mock controls, but the authors zoom in on ISC cells with very low AWD expression in the neighboring cells. I wonder why. The HA expression levels in some ISC cells of awd knockdown in Fig. 3d are high, but the authors zoom in on ISCs with low HA staining. Is there a reason in choosing representative results?

To make it easy for readers to understand the results, I suggest the region of interest is put into the center rather than at the edge of the panel (Fig. 1e, 2d, 3a, 3b etc.).

Minor points:

Although MARCM technique is well known, the author should describe the exact genotypes for MARCM analysis in methods or figure legends.

Nowhere to find the logic for using Arm+Prospl labeling in Fig. 2d and the full name for AHS (Fig. 5d). More background information is needed to fully understand the results.

The authors only show the immunostaining evidence to conclude that pMAD expression is altered in Fig 5a-c. Is it possible to perform Western analysis to confirm the finding?

Does overexpression of constitutively active Tkv (UAS-TKVca) rescue the phenotype in Fig 6c?

I would prefer AWD-facilitated rather than AWD-mediated endocytosis of Tkv.

Reviewer #3 (Remarks to the Author):

The authors present data on the effects of Dpp signaling in intestinal stem cells in the *Drosophila* midgut. There has been some controversy about the role of this pathway in this system, with several studies generally suggesting it exerts a pro-proliferative effect during regeneration, but others showing an anti-proliferative effect. The authors build on their previous study that suggested the initial pro-proliferative regeneration phase is driven by the Dpp receptor Sax, and the subsequent anti-proliferative phase is driven by another receptor, Tkv. The current study sheds light on the molecular mechanism underlying the Sax-to-Tkv switch. Tkv ubiquitination is present during the initial stages of regeneration to promote proliferation through Sax, which is followed by activation of Tkv through JNK pathway, and the internalization of membrane associated Tkv by AWD to promote the second anti-proliferative phase of Dpp signaling. The model presented is actually quite compelling and the data is of high quality. The following issues should be addressed for publication:

1. What happens to Sax during the later phases of regeneration? It is still in the cell presumably, but must either compete with Tkv for Dpp signaling, or is deactivated or degraded... Do the levels of Tkv raise so high as to outcompete Sax? The role of Sax which drives the proliferative/regenerative response is unclear during times when Tkv is active. This raises the possibility that pro- and anti-proliferative Dpp signaling compete during the transition between proliferation and return to quiescence.
2. What is the timeline of this process? The authors usually examined stem cells ~18h after Ecc15 infection, but it is not clear if the AWD-Tkv-Dpp pathway is present during the earlier phases of stress. The reason this is relevant is because the initial response to Dpp should be to increase proliferation, rather than suppress it due to Tkv activity, so it should follow that a lag time is present in the first phase of the regenerative response. The authors show a nice test of the proliferation timeline in Fig 6a, and a similar experiment showing the Highwire, AWD, and Tkv changes compared to the proliferation phase would be informative.
3. Somewhere in this manuscript (intro or discussion) the authors should mention the Hippo pathway. It too, in part, accounts for the difference between stem cell quiescence and regenerative activity, as repressors of Yki are active in the former but absent in the latter. This pathway is likely playing a parallel role to Dpp in this system, but is not mentioned. A paragraph discussing the relationship of the proposed model to this and/or other pathways involved in regeneration would be appropriate in the discussion (does not have to be exhaustive, since there are many processes occurring).
4. The concluding sentence of the abstract contains the words: "critical", "new", "importance". These words are superfluous and could be removed.

Response to reviewers -

We would like to thank the reviewers for the positive reception of our work and for the constructive critiques. We appreciate that the reviewers found the biological questions addressed of considerable potential significance, the study exciting, our model compelling, and our data of high quality. Nevertheless, the reviewers also raised important concerns regarding further quantitative evidence for the reported changes of Highwire, Tkv and phospho-Mad levels, Highwire - mediated regulation of Tkv, the role of Sax in the later phase of regeneration, as well as the timeline of the biphasic regenerative response. We have now performed additional experiments to address these questions and have obtained new data that strengthen our manuscript and, we believe, fully satisfy the reviewers' concerns. We summarize the new data here and respond to the reviewers' comments point-by-point below.

Reviewers 1 and 2 recommended western blot analysis to confirm the reported changes in expression of various proteins under the studied conditions in ISCs. We agree that additional quantitative evidence in addition to the reported immunohistochemistry data would strengthen our conclusions. However, western blots are not applicable in this study, as it relies on characterization of protein expression in individual ISCs rather than bulk tissue, and western blots on purified ISCs would require dissection and FACS sorting of thousands of guts for each data point. We have therefore sought an alternative method, and applied a flow cytometry method that would allow us to measure protein expression in individual ISCs purified from guts in the corresponding conditions. We have now performed such analysis for the expression of Tkv, Sax, Highwire and Awd in ISCs in time-courses following *Ecc15* infection as well as to quantify the levels of phospho-MAD in specific conditions. This new analysis not only clarifies the timeline of the biphasic regenerative response, but also addresses the concerns of reviewer 3, as discussed in the point-by-point response.

Reviewer 1 requested direct evidence for the necessity of Highwire's ubiquitin ligase activity for Tkv degradation. We agree that this is an important question, yet have the same problem of limited input material as described for the western blots above. It is thus not possible to detect ISC-specific ubiquitination of Tkv using biochemical assays. However, we used a genetic approach to address this question, and found that overexpression of Highwire carrying two mutations that specifically disrupt its E3 ubiquitin ligase activity (UAS::*highwire*^{ΔRING}) causes ectopic induction of Tkv in ISCs, consistent with our model. We have also carried out flow cytometry quantification to demonstrate the accumulation of Tkv in ISCs in these conditions. We believe that these new data demonstrate that the E3 ubiquitin ligase activity of Highwire is required for Tkv degradation.

In addition to these new experiments, we have also carried out additional experiments recommended by the reviewers, including qPCR analysis of *tkv* transcription in ISCs, quantification of Tkv subcellular localization based on single confocal Z slices, improved resolution for pMAD and Awd staining, additional statistical analysis throughout the manuscript, as well as additional clarifications for antibody use in figure legends and annotations for fly genetic background. Experiments addressing other minor concerns have also been included, as discussed in the detailed response.

Altogether, we believe these new data result in substantial improvement of the manuscript, addressing the most significant concerns of the reviewers, and supporting our proposed model.

Specific Responses (Bold)

Reviewer #1 (Remarks to the Author):

In *Drosophila*, intestinal regeneration induced by injury involves two steps: 1) early activation of intestinal stem cell (ISC) proliferation and 2) subsequent return to homeostatic status. It has been shown previously that differential regulations of the BMP type I receptor Tkv is critical for these two steps of the regenerative process, however, the underlying molecular mechanism remains unclear. In the manuscript, Cai et al showed that the E3 ubiquitin ligase Highwire and proteasome maintain Tkv protein at low levels during homeostatic condition. They further found that upon *Ecc15* infection, Tkv was temporarily stabilized and internalized into Rab5-positive endocytic vesicles. Moreover, they found that the internalization of TKV in ISCs is regulated by AWD, which is upregulated by *Ecc15* infection. Finally, the authors performed extensive experiments to show that the JNK/AWD/TKV regulatory axis is involved in the re-establishment of ISC quiescence.

Overall, this work are of potential interest and would provide insights in understanding how the precise regulation of Tkv contributes to intestinal regeneration. It is worth to note that the manuscript includes two parts: 1) describing the role of Highwire and proteasome in maintaining Tkv protein at a low level under homeostatic condition and 2) investigating functional contribution of AWD to Tkv internalization and the re-establishment of ISC quiescence. However, two parts don't come into together in a cohesive way, and interpretations of some data are confusing. Thus, additional biochemistry experiments elucidating the molecular mechanisms of how TKV stability is regulated by Highwire in homeostatic condition and how Tkv is activated by AWD during internalization are necessary in a revised manuscript.

Below are specific comments:

1. Regulation of Tkv by Highwire is not clear, since all conclusions were made from immunostaining experiments, levels of endogenous TKV mediated by Highwire and proteasome in guts under homeostatic or infective conditions should be examined by western blotting assays.

We agree with the reviewer that additional quantitative assays would significantly support our model for the post-transcriptional regulation of Tkv by Highwire and the proteasome. However, western blots using whole gut tissues would not be informative in this context, as the effects we describe are ISC-specific and Tkv expression in other cell types of the gut would be masking any effects in ISCs. Furthermore, collecting ISCs by FACS does not provide sufficient material for western blots. For these reasons, we have applied a Flow Cytometry assay to quantify TkvHA levels in freshly purified ISCs (Fig. 1f, 2e, 2e'). Our data show that feeding flies with 20uM proteasome inhibitor PS341 for 2 days, knocking down Highwire, or disrupting E3 ubiquitin ligase activity of Highwire specifically in ISCs by overexpressing full length Highwire with two mutations in its RING finger domain (UAS-*hiw*^{ΔRING}) significantly upregulates TkvHA levels (Fig.2e, 2e'), consistent with immunostaining experiments. In addition, time-dependent induction of TkvHA in ISCs following *Ecc15* infection was also confirmed by flow cytometry analysis (Fig.1f).

2. As seen from several figures, levels of Tkv were strongly induced in guts upon infection of *Ecc15* or by overexpression of AWD. Authors explained that these changes were mainly attributed to the inhibition of proteasome activity. The transcriptional reporter of TkvA used in the work is a transgene that may not fully recapitulate the behavior of endogenous tkv. The reviewer suggest

that authors perform qPCR to measure levels of *tkv* transcript under the investigated conditions in a time-course way.

We agree, and have performed qPCR analysis of *tkv* transcription in purified ISCs in a time-course following *Ecc15* infection (Fig.1c). Our results show that *tkv* mRNA level is not significantly changed during the later phase of regeneration (i.e. 12h or 18h post-*Ecc15* infection) although Tkv protein is significantly induced during that time, further supporting our current conclusion that Tkv is regulated post-transcriptionally in ISCs.

Based on the reviewer's comment, we believe that we also needed to resolve a slight misunderstanding of our model: We propose that in the later phase of the regenerative response, Tkv is both stabilized and internalized in ISCs, and that these are two separate steps: the stabilization of Tkv is due to the downregulation of proteasome activity in ISCs, and the internalization of Tkv is mediated by Awd, as shown in Fig.3 and Fig.S3. We do not propose that overexpressing Awd alone would stabilize TkvHA in ISCs. In fact, we observed that TkvHA is not detectable in ISCs overexpressing Awd without *Ecc15* infection, suggesting that Awd can only promote TkvHA internalization after it is stabilized. We have now incorporated these data in the supplementary (Fig. S3a), and edited the discussion to clarify our interpretation of these results.

3. In abstract, authors claimed that "Tkv protein levels are kept low during homeostasis by the activity of the E3 ubiquitin ligase Highwire and degradation by the ubiquitin-proteasome system". However, authors did not provide any evidence showing that Highwire targets and regulates Tkv through ubiquitination and that enzymatic activity is required for Tkv ubiquitination. The reviewer suggest that authors measure levels of ubiquitinated Tkv in guts with/without knockdown of highwire and in guts treated with/without proteasome inhibitor PS-341. Since the authors claimed "Inhibiting proteasome function by feeding flies the proteasome inhibitor PS-341 for 2 days resulted in similar ectopic expression of Tkv in ISCs without activation of pMAD (Fig. 2c)", the reviewer want to see if there is difference in levels of ubiquitinated Tkv between gut samples with highwire knockdown and samples treated with proteasome inhibitor.

We agree with the reviewer that demonstrating that ubiquitin ligase activity of Highwire is required for the turnover of Tkv is required to support the conclusion of the paper. Importantly, Highwire has been well characterized as an E3 ubiquitin ligase, and we had already shown that either loss of Highwire (by RNAi) in ISCs, or the introduction of a deletion of its N-terminal domain(Hiw^{ΔN}) that includes the RING finger domain that is required for its E3 ubiquitin ligase activity, causes ectopic induction of Tkv under homeostatic conditions.

To further support our model, we have now overexpressed specifically in ISCs a full length Highwire with two mutations that have been reported to specifically disrupt its E3 ubiquitin ligase activity (UAS::*hiw*^{ΔRING}, Wu, C., *et al.* 2005). This also resulted in ectopic induction of Tkv in ISCs (Fig.2c,2c,2e,2e'), consistent with our previous data. This finding further supports the idea that the E3 ubiquitin ligase activity of Highwire is required for Tkv degradation under homeostatic conditions.

We agree that biochemical data showing TkvHA ubiquitination *in vivo* in ISCs would be further supporting this model. However, technical limitations preclude us from performing these experiments. Even immunoprecipitating an over-expressed Tkv-GFP from ISCs from

300 dissected guts with GFP antibody did not result in sufficient material to obtain clear signals on a western blot.

4. Since Highwire and AWD were proposed to regulate Tkv dynamics during homeostasis and early ISC activation, respectively, the major question the reviewer wants to ask is what relationship between Highwire and AWD is. It has been proposed that the receptor internalization is often associated with its ubiquitination. Given that AWD promotes Tkv internalization, thus possibly promoting Tkv ubiquitination, the question becomes as to whether Highwire contributes to the AWD-mediated Tkv internalization. Authors should discuss this issue, if Highwire has no role in the later stage of gut regeneration.

This is an interesting suggestion. In fact, our results (Fig. S3c) indicate that in this case, loss of highwire does not prevent the internalization of TkvHA when Awd is over-expressed. We thank the reviewer for pointing this out and we have now included this discussion in the main text.

Reviewer #2 (Remarks to the Author):

The study entitled "AWD regulates BMP Signaling in intestinal stem cells to maintain tissue homeostasis" identifies Highwire-mediated Tkv degradation and AWD mediated internalization of Tkv as essential steps for the activation of MAD signaling in ISCs. They further link this mechanism to JNK-mediated activation of AWD expression, thus providing a model for the transition from activated ISCs to resting ISCs after regeneration has concluded. This is an exciting story elucidating relevant pathways regulating ISC and tissue homeostasis. While the manuscript and the experiments described therein provide a wealth of experimental data, critical evidence is still missing to support major conclusion and many important questions remain.

We would like to thank the reviewer for the positive reception of our work. We believe that our new data and edits to the manuscript address the reviewer's concerns and have strengthened the manuscript substantially.

Major comments

1, Lack of statistical analysis for critical results:

Lack of quantification in Fig. 1b, Fig. 1c, Fig. 2a-c, Fig. 4a-d, and Fig. 5a-c.

This reviewer is not convinced by the data shown that Tkv protein, but not transcripts of tkv, is affected by *Ecc15* infection.

We thank the reviewer for pointing this out. We have now included quantifications for all the noted figures as well as for supplemental figures. We have now also performed qPCR analysis of Tkv transcription in FACS purified ISCs in a time-course following *Ecc15* infection. The results of these experiments are consistent with our results using the Tkv transcriptional reporter, and confirm that Tkv is not regulated at the transcriptional level in ISCs during the regenerative response (Fig.1c).

Co-localization of Rab5, Tkv, and Lysosome in Fig. 1f also needs to be quantified based on images from single slice or a few slice projections. Co-localizations analysis on images from thick tissues could lead to false positive results. LysoTracker labels acidic organelles such as lysosome; this might explain the co-localization between lysosome and the Rab5-positive early endosomes.

We have now replaced representative images in Fig. 1f with single-slice images, and incorporated quantifications of TkvHA subcellular localization in different vesicular compartments, based on single-slice images.

2, Image quality needs to be improved throughout. For example, the resolution for anti-pMAD staining is too low in Fig. 2a and 2c. The anti-pMAD staining pattern in controls (control RNAi and mock-treated) is greatly different without any explanation.

We have now improved the quality of pMAD staining throughout the paper with higher resolution images, including Fig. 2a and 2c.

Inconsistent patterns of anti-HA staining are also noted in Fig. 1C and 1F in similar conditions.

We thank the reviewer for pointing this out. The staining patterns of TkvHA in Fig.1c and Fig.1f are different, as two antibodies from different species were applied respectively due to different experimental requirements. In Fig. 1c, we had to use rabbit anti-HA antibody to co-stain with rat anti-Delta antibody, confirming Tkv is expressed in ISCs. We have tried three different rabbit anti-HA antibodies (Novus Biologicals NB600-363, Cell Signaling 3724S and Abcam ab137838), but all of them exhibited a lot of background. For any other experiments we therefore used rat anti-HA antibody, which results in immunostaining with a better signal-to-noise ratio. We have now annotated such differences in the legends and materials.

Proper controls are needed in many places. For example, Delta immunostaining is needed to label ISC in Fig. 3e, as used in Fig. 4e and 4f.

In Fig. 3e, we show differences of TkvHA localization inside and outside of *awd* mutant MARCM clones. As shown in Fig. S4a, all the cells inside of *awd* mutant clones are positive for Delta staining (now quantified). Since anti-HA and anti-DI antibodies are not compatible, we could not directly perform double-staining in these clones, but we believe that these figures still clearly demonstrate the differential intracellular localization of TkvHA in AWD-deficient vs wt cells.

There is obvious AWD expression in GFP+ and GFP- cells in Fig 3a Mock controls, but the authors zoom in on ISC cells with very low AWD expression in the neighboring cells. I wonder why.

In Fig.3a, we focused on comparing AWD expression in GFP+ cells of mock controls to infected intestines. The background AWD staining pointed out by the reviewer is staining in the visceral muscle, which is in close proximity to the ISCs. We have now quantified the levels of anti-AWD immunofluorescence.

The HA expression levels in some ISC cells of *awd* knockdown in Fig. 3d are high, but the authors zoom in on ISCs with low HA staining. Is there a reason in choosing representative results?

We think the slightly higher HA expression in some cells of *awd* knockdown in Fig.3d that the reviewer pointed out are not in ISCs. Those cells with higher HA staining are GFP-, thus not the cells of interest to zoom in. Critically, we are not making a statement regarding expression levels of TkvHA in this case, but the presence or absence of TkvHA+ puncta (i.e. vesicles).

To make it easy for readers to understand the results, I suggest the region of interest is put into the center rather than at the edge of the panel (Fig. 1e, 2d, 3a, 3b etc.)

We thank the reviewer for pointing this out. We adjusted the regions of interest as requested.

Minor points:

Although MARCM technique is well known, the author should describe the exact genotypes for MARCM analysis in methods or figure legends.

We have now added the genotype for MARCM clone analysis in the methods.

Nowhere to find the logic for using Arm+Prospero labeling in Fig. 2d and the full name for AHS (Fig. 5d). More background information is needed to fully understand the results.

We are sorry for the lack of information. The Armadillo antibody (Arm) labels cell membranes and shows elevated signal in ISC/EB nests, while Prospero antibody (Prospero) labels enteroendocrine cells (EEs). Co-staining with Delta and Arm+Prospero helps identify ISCs even if the GFP-CL1 signal is low and was thus used to quantify GFP-CL1 signal in ISCs. AHS stands for 'after heat shock'. We have now noted this information in the figure legend.

The authors only show the immunostaining evidence to conclude that pMAD expression is altered in Fig 5a-c. Is it possible to perform Western analysis to confirm the finding?

We agree with the reviewer that additional quantitative assays would significantly support our data in Fig. 5a-5c. However, western blots of whole gut tissues would not provide specific information on pMAD activity in ISCs, as pMAD can also be detected in enterocytes (ECs) and other cells. We were further not able to obtain enough FACS sorted ISCs for informative western blots. Instead, we performed flow cytometry analysis of pMAD levels in freshly purified ISCs (Fig. 5b, 5b, 5d, 5d', 5f, 5f'), which confirmed our original findings (now in Fig.5a, 5c, 5e).

Does overexpression of constitutively active Tkv (UAS-TKV^{ca}) rescue the phenotype in Fig 6c?

We have now included the rescue experiments measuring ISC mitotic activity, gut barrier function and host resistance to acute infection with constitutively active Tkv (UAS::tkv^{QD}). Our data show that overexpression of Tkv^{QD} can rescue ISC over-proliferation caused by *awd* loss of function upon 24h *Ecc15* infection (Fig. 6d), while gut permeability and fly survival in response to acute *Pe* infection could not be rescued (Fig. 6f, 6h). In fact, we found that Tkv^{QD} overexpression alone significantly inhibits ISC proliferation (Fig. S6c) and impairs the host's capability of resisting acute *Pe* infection (Fig. 6h), indicating that overly inhibiting ISC proliferation with UAS::tkv^{QD} substantially impairs gut epithelium regeneration. We have now included all these new results in the main text.

I would prefer AWD-facilitated rather than AWD-mediated endocytosis of Tkv.

We have now adjusted the text.

Reviewer #3 (Remarks to the Author):

The authors present data on the effects of Dpp signaling in intestinal stem cells in the *Drosophila* midgut. There has been some controversy about the role of this pathway in this system, with several studies generally suggesting it exerts a pro-proliferative effect during regeneration, but others showing an anti-proliferative effect. The authors build on their previous study that suggested the initial pro-proliferative regeneration phase is driven by the Dpp receptor Sax, and the subsequent anti-proliferative phase is driven by another receptor, Tkv. The current study sheds light on the molecular mechanism underlying the Sax-to-Tkv switch. Tkv ubiquitination is present during the initial stages of regeneration to promote proliferation through Sax, which is followed by activation of Tkv through JNK pathway, and the internalization of membrane associated Tkv by AWD to promote the second anti-proliferative phase of Dpp signaling. The model presented is actually quite compelling and the data is of high quality. The following issues should be addressed for publication:

1. What happens to Sax during the later phases of regeneration? It is still in the cell presumably, but must either compete with Tkv for Dpp signaling, or is deactivated or degraded... Do the levels of Tkv raise so high as to outcompete Sax? The role of Sax which drives the proliferative/regenerative response is unclear during times when Tkv is active. This raises the possibility that pro- and anti-proliferative Dpp signaling compete during the transition between proliferation and return to quiescence.

We would like to thank the reviewer for the positive remarks on the significance of our study. We showed that Sax is continuously expressed in ISCs under homeostatic conditions and after infection (Fig. S1f). We confirmed this using flow cytometry analysis of Sax levels in ISCs during the time-course of *Ecc15* infection for 24h (Fig. S1g). The flow cytometry shows a slight (less than 2-fold) up-regulation of Sax in ISCs during the regenerative episode, coinciding with the (stronger) increase of Tkv expression (Fig. 1e, 1f). This finding not just confirms that Sax is not degraded during the recovery phase when Tkv is induced, but also excludes the possibility that the return to ISC quiescence is caused by loss of Sax.

Our previously published study (Ayyaz, *et al.* 2015) had shown that nuclear localization of Smox, which acts downstream of Sax, is seen in ISCs during the early proliferative phase but lost in the later recovery phase, suggesting that Sax/Smox pro-proliferative signaling is not active during the late recovery phase. This study also showed that inhibiting ISC proliferation by knocking down Sax does not prevent the induction of Tkv at 24h post-*Ecc15* infection, indicating that Sax is not required for Tkv expression. In addition, we have now included the new result that forced Sax over-expression cannot competitively prevent Tkv induction during the later phase of regeneration (Fig. S1e, S1e'), suggesting that the presence of Tkv in ISCs provides a dominant signal that inhibits ISC proliferation. Our interpretation of these results is that Tkv can compete out Sax from Sax/Punt heterotetramers, resulting in predominant Tkv/Punt heterotetramers and activation of MAD (as also described in Haerry *TE.* 2010). We are now including an extended discussion regarding the relative roles of Sax and Tkv in the regulation of ISC proliferation.

2. What is the timeline of this process? The authors usually examined stem cells ~18h after *Ecc15* infection, but it is not clear if the AWD-Tkv-Dpp pathway is present during the earlier phases of stress. The reason this is relevant is because the initial response to Dpp should be to increase proliferation, rather than suppress it due to Tkv activity, so it should follow that a lag time is present

in the first phase of the regenerative response. The authors show a nice test of the proliferation timeline in Fig 6a, and a similar experiment showing the Highwire, Awd, and Tkv changes compared to the proliferation phase would be informative.

We thank the reviewer for this suggestion. We have now, in addition to our previous time-course analysis of TkvHA expression based on immunostaining, also quantified Tkv, Highwire, and Awd level in ISCs during a time-course of infection for 24h using flow cytometry of FACS-purified ISCs, and found that the peak of Awd upregulation is around 12h after *Ecc15* infection (Fig. 3b), while that of Tkv happens at 24h post infection (Fig. 1f). Even though Highwire level is found to be slightly upregulated around 12h, its expression wasn't significantly changed in all other time points (Fig. 2g). In summary, our findings from flow cytometry assays further support the proposed model in Fig. 7 that induction of Tkv is not a consequence of Highwire downregulation (but rather likely due to proteasome inhibition), and that Awd induction in the early phase ensures its later role in mediating Tkv internalization during the recovery phase. We have now proposed a relationship between relative expression of Tkv, Highwire, Awd and Sax with ISC proliferation during one regenerative episode in Fig.7, based on these data.

3. Somewhere in this manuscript (intro or discussion) the authors should mention the Hippo pathway. It too, in part, accounts for the difference between stem cell quiescence and regenerative activity, as repressors of Yki are active in the former but absent in the latter. This pathway is likely playing a parallel role to Dpp in this system, but is not mentioned. A paragraph discussing the relationship of the proposed model to this and/or other pathways involved in regeneration would be appropriate in the discussion (does not have to be exhaustive, since there are many processes occurring).

We have now mentioned Hippo signaling in the discussion and cited additional studies to ensure we are describing the signaling pathways influencing ISC proliferation accurately and comprehensively. Due to space limitations, we can't go into too much detail about these pathways, but our interpretation of the literature is that Yki activation results in the activation of ISC proliferation through both cell-autonomous (Yki activation in ISCs) and non-autonomous (Yki activation in ECs, which results in elevated Upd expression) mechanisms. We are not aware of data that reveal a specific mechanism by which Yki repression mediates the return to ISC quiescence, but we would be happy to include a discussion of this if the reviewer points out the specific study to cite.

4. The concluding sentence of the abstract contains the words: "critical", "new", importance". These words are superfluous and could be removed.

We have now adjusted the text as suggested.

REVIEWERS' COMMENTS:

Reviewer #1 (Remarks to the Author):

The authors have addressed my questions, and I would support publication of the manuscript in Nature Communications.

Reviewer #2 (Remarks to the Author):

The authors have addressed all of my concerns and questions.

Reviewer #3 (Remarks to the Author):

The authors have addressed my comments and questions.

We would like to thank all three reviewers for their positive reception of our work, and we are glad that they all support publication of the reported findings in *Nature Communications*.

REVIEWERS' COMMENTS:

Reviewer #1 (Remarks to the Author):

The authors have addressed my questions, and I would support publication of the manuscript in Nature Communications.

Reviewer #2 (Remarks to the Author):

The authors have addressed all of my concerns and questions.

Reviewer #3 (Remarks to the Author):

The authors have addressed my comments and questions.